# Addressing Node Integration Skewness in Graph Neural Networks Using Hop-Wise Attention

## Abstract

Graph neural networks (GNNs) often suffer performance degradation as their layer count grows, typically due to the well-known problems of over-smoothing and over-squashing. In this work, we identify an additional factor contributing to this degradation, which we term the K-skewed-traversal problem: certain hop distances are disproportionately emphasized during aggregation, with this emphasis intensifying as the number of layers grows. To address this, we introduce an algorithm called Hop-wise Graph Attention Network (HGAT) that ensures uniform aggregation across hops to eliminate the K-skewed-traversal problem, and employs a hop-wise attention mechanism to adaptively prioritize specific hop distances. We theoretically prove that HGAT removes this skewness by balancing contributions from different hop distances before applying hop-wise attention. Moreover, in our extensive empirical evaluation[1], we observe notable improvement in terms of solution quality compared to the state-of-the-art GNN models, particularly as the number of layers increases.

## 1 Introduction

Graph neural networks (GNNs) (Gori et al., 2005; Scarselli et al., 2009) have become increasingly popular because they can effectively model graph-structured data, capturing complex relationships between nodes. These networks can model many real-life interactions, information, or data across various fields such as social networks (Fan et al., 2019), biological networks (Bongini et al., 2023), chemical systems (Wu et al., 2023), multi-agent systems (Kortvelesy et al., 2023), etc. The advent of GNNs has accelerated the exploitation of complex graph data structures, leading to significant advancements in these areas.

Currently, several methods exist in the GNN literature, including GCN (Kipf & Welling, 2017), GraphSAGE (Hamilton et al., 2017b), and GAT (Veličković et al., 2018). They are the pioneers of the message-passing GNN. They are called message-passing GNN because all of these methods share a common mechanism: each node integrates its neighbors' features with its own to capture local patterns, thereby leveraging the graph's structure for feature learning (Xie et al., 2020). They differ only in how they perform this integration.

It is worth noting that most GNN problems require interactions not only between a node and its immediate neighbors but also between nodes that are not directly connected. For example, in molecular graphs, a chemical property can depend on the interaction of atoms located on opposite sides of the molecule (Ramakrishnan et al., 2014; Gilmer et al., 2017), necessitating distant integration across the graph. To facilitate this, GNNs integrate features from both neighboring and distant nodes by stacking several layers of the network. The more layers stacked, the broader the range of information a node can integrate. However, observations show that GNNs do not learn much from more than a few layers, a limitation we refer to as the problematic-radius problem. The well-demonstrated reasons for the problematic-radius problem are over-smoothing (Li et al., 2018; Chen et al., 2020; Wu et al., 2021) and over-squashing (Alon & Yahav, 2021; Di Giovanni et al., 2023). Over-smoothing occurs when node representations become indiscernible as the number of layers increases. Also, as the number of layers increases, the number of nodes from where a node integrates information grows exponentially. This exponentially growing information is summarized into a fixed-length vector, causing over-squashing. As a result, a node receives weaker information from distant nodes, and stronger information from nearby nodes.

---

[1]The implementation is available at `https://drive.proton.me/urls/XSGJ8SJJGW#ObIyDVkZDqTi`

Beyond over-smoothing and over-squashing, we identify another problem in traditional GNNs that contributes to the problematic-radius problem. We call this the K-skewed-traversal problem. This problem arises by stacking several GNN layers to integrate information from multiple hops. Depending on their sub-graph structures, GNNs can randomly assign different priorities to nodes. For instance, a GNN might prioritize a node's second-hop neighbors over its first-hop neighbors, while another node might experience the opposite priority. Also, this skewness among a node's neighbors' priorities grows with the increase in hop counts. As a consequence of these arbitrary and increasingly skewed priorities, GNNs struggle to learn common patterns among nodes, and consequently, adding more layers yields little benefit.

Notably, the K-skewed-traversal problem is not a deliberate design choice, but rather an unintended byproduct of the standard aggregation process in models like GCN, GraphSAGE, and GAT. For example, in GCN and GraphSAGE, this problem arises from unintentional biases in their aggregation processes. In GAT, which introduces attention, the issue manifests even before the attention parameters have been learned, since the initial node representations are already skewed when the attention weights are applied. These biases are not explicitly encoded rules or optimizations; rather, they intrinsically arise from the normal computation in traditional GNNs.

In light of the above background, this paper aims to demonstrate the susceptibility of traditional GNNs to the K-skewed-traversal problem, especially in deep architectures. We analyze the origins of this issue and empirically demonstrate its impact on node representations. To address this, we propose a novel algorithm, the Hop-wise Graph Attention Network (HGAT), which mitigates the K-skewed-traversal problem by ensuring that nodes from all hops receive equal attention. Additionally, HGAT incorporates a hop-wise attention mechanism that adaptively prioritizes specific hop distances, thereby enhancing the network's ability to learn informative and balanced representations from neighbors across multiple layers. Finally, we empirically demonstrate that HGAT outperforms state-of-the-art GNNs in node classification accuracy, particularly as the number of layers increases.

The remainder of this paper is structured as follows. We provide a detailed description of the K-skewed-traversal problem with worked examples in the section that follows. Section 3 details the proposed HGAT methodology. Next, we offer the theoretical analysis in Section 4. We then present the empirical results of our method compared to the state-of-the-art in Section 5. Finally, Section 6 concludes.

## 2 Problem Formulation

In this section, we introduce a Generalized GNN Framework (GGNNF), which unifies the mechanisms of GCN, GraphSAGE, and GAT. We then highlight the K-skewed-traversal problem that arises from this framework and proceed to mathematically formulate the issue.

### 2.1 Generalized GNN Framework (GGNNF)

Motivated by the representation generation algorithm used in GraphSAGE, we propose a Generalized GNN Framework (GGNNF) to unify the representation generation mechanism of the state-of-the-art GNN methods GCN, GraphSAGE, and GAT. Although these methods vary slightly in their approaches, they share a common underlying procedure: each node's representation is generated by aggregating information from its neighbors' previous representations and concatenating this with its own previous representation. Here, the representation of a node is a mapping of that node to a point in vector space, encoding the structural and feature-based information of the original node. This shared procedure allows us to generalize their mechanisms, as shown in Algorithm 1.

Algorithm 1 takes as input a graph $\mathcal{G}(\mathcal{V}, \mathcal{E})$ and the feature vectors $\mathbf{x}_v$ of all nodes $v \in \mathcal{V}$. The algorithm begins by initializing each node's representation $\mathbf{h}_v^0$ with its input feature $\mathbf{x}_v$ (Line 1). The outer loop (Line 2) iterates over the layers from 1 to $K$, where $K$ represents the farthest hop distance from which a node gets information. Within this loop, each node $v$ aggregates information from its first-hop neighbors $\mathcal{N}(v)$ using an aggregation function $AGGREGATE_k$, producing the aggregated neighborhood vector $\mathbf{h}_{\mathcal{N}(v)}^k$ for the $k$-th layer (Line 4). This aggregated neighborhood information is concatenated with the node's previous representation $\mathbf{h}_v^{k-1}$ using the $CONCAT$ function. GraphSAGE proposes several aggregator architectures for $AGGREGATE_k$ —such as mean, LSTM, and pooling—and uses literal vector concatenation for $CONCAT$. In GGNNF, we treat these as abstract placeholders, not fixed operations, allowing them to be defined as needed.The concatenated vector is then passed through a fully connected layer, where it is multiplied by

---

**Algorithm 1** Representation generation of GGNNF

---

**Require:** Graph $\mathcal{G}(\mathcal{V}, \mathcal{E})$; input features $\{\mathbf{x}_v, \forall v \in \mathcal{V}\}$; depth $K$; weight matrices $\mathbf{W}^k, \forall k \in \{1, \ldots, K\}$; non-linearity $\sigma$; differentiable aggregator functions $AGGREGATE_k, \forall k \in \{1, \ldots, K\}$; neighborhood function $\mathcal{N} : v \to 2^{\mathcal{V}}$

**Ensure:** Vector representations $\mathbf{z}_v$ for all $v \in \mathcal{V}$

1: $\mathbf{h}_v^0 \leftarrow \mathbf{x}_v, \forall v \in \mathcal{V}$;
2: **for** $k = 1 \ldots K$ **do**
3:     **for** $v \in \mathcal{V}$ **do**
4:         $\mathbf{h}_{\mathcal{N}(v)}^k \leftarrow AGGREGATE_k(\{\mathbf{h}_u^{k-1}, \forall u \in \mathcal{N}(v)\})$
5:         $\mathbf{h}_v^k \leftarrow \sigma(\mathbf{W}^k \cdot CONCAT(\mathbf{h}_v^{k-1}, \mathbf{h}_{\mathcal{N}(v)}^k))$
6:     **end for**
7:     $\mathbf{h}_v^k \leftarrow \mathbf{h}_v^k / \left\|\mathbf{h}_v^k\right\|_2$
8: **end for**
9: $\mathbf{z}_v \leftarrow \mathbf{h}_v^K, \forall v \in \mathcal{V}$

---

a weight matrix $\mathbf{W}^k$, followed by a non-linear activation function $\sigma$, which generates the node's updated representation $\mathbf{h}_v^k$ for the current layer (Line 5). The new representation is then normalized (Line 7). After all $K$ layers have been processed, the final output $\mathbf{z}_v$ is the representation of node $v$, computed for all nodes in the graph (Line 9).

Each GNN method in GGNNF aggregates neighborhood features and concatenates them with the node's own features, but the aggregation mechanisms differ slightly across models. Below, we outline how each model fits into the GGNNF framework:

**GCN:** The $AGGREGATE_k$ and $CONCAT$ operations are both replaced by summation, leading to the following simplified update rule, which replaces Lines 4 and 5:

$$\mathbf{h}_v^k \leftarrow \sigma(\mathbf{W}^k \cdot \frac{\sum_{u \in \mathcal{N}(v) \cup v} \mathbf{h}_u^{k-1}}{|\{v\} \cup \mathcal{N}(v)|})$$

**GraphSAGE:** Rather than aggregating all neighbors, GraphSAGE randomly samples a subset of neighbors for each node (uniformly at random) and aggregates only those neighbors' features. This modification replaces Line 4 with:

$$\mathbf{h}_{\mathcal{N}(v)}^k \leftarrow AGGREGATE_k(\{\mathbf{h}_u^{k-1}, \forall u \in \{u_1, u_2, u_3, \ldots, u_l \overset{i.i.d}{\sim} Uniform(\mathcal{N}(v))\}\})$$

Here, $l$ represents the size of the independent random sample taken from the neighborhood of node $v$.

**GAT:** Every node and its neighborhood nodes' features are aggregated according to a learned attention value $\alpha_{vu}$, independently specified for every pair of the nodes $u$ and $v$, where $v$ is the node of interest and $u$ is a node in its neighborhood. To stabilize the learning process, multi-head attention (Vaswani et al., 2017) is applied, where $M$ such independent attention mechanisms are executed in parallel (Veličković et al., 2018). This technique allows the model to capture diverse relational aspects by combining multiple attention scores. Thus, Lines 4 and 5 are replaced with the following statement:

$$\mathbf{h}_v^k \leftarrow \sigma(\frac{1}{M} \sum_{m=1}^M \sum_{u \in \mathcal{N}(v) \cup v} \alpha_{vu}^m \mathbf{W}^{km} \mathbf{h}_u^{k-1})$$

Here, $\alpha_{vu}^m$ represents the $m$-th attention head's learned attention value between the node of interest $v$ and a neighboring node $u$, while $\mathbf{W}^{km}$ denotes the $k$-th weight matrix for the $m$-th attention head.

The shared procedure among GCN, GraphSAGE, and GAT enables us to unify them under the GGNNF. This framework allows for a comprehensive investigation of all three methods, as we discuss in the following section.

## 2.2 The K-skewed-traversal Problem

We now explain how the GGNNF described above results in the K-skewed-traversal problem. We categorize GCN and GAT as deterministic GGNNF models (because of fixed neighborhood aggregation) and Graph-SAGE as a randomized GGNNF model (due to its random sampling in neighborhood aggregation). We analyze these two cases separately.

### 2.2.1 Deterministic GGNNF

To illustrate how GGNNF generates the nodes' representation of a graph, let's consider the graph depicted in Figure 1a. To capture interactions from nodes that are, for example, two hops away, we stack two layers of a deterministic GGNNF, generating representations $\mathbf{h}_v^2$ for every node $v$. Focusing on node $\mathbf{a}$, its representation $\mathbf{h}_a^2$ is computed using its neighbor nodes' previous layer representation ($\mathbf{h}_b^1$ and $\mathbf{h}_c^1$), along with its own previous layer representation ($\mathbf{h}_a^1$). Node $b$'s representation $\mathbf{h}_b^1$, in turn, depends on its neighbor nodes' previous layer representation ($\mathbf{h}_a^0$ and $\mathbf{h}_d^0$) and its own previous layer representation ($\mathbf{h}_b^0$), and so on. We can visualize this iterative process through a computation graph, motivated by Hamilton et al. (2017a), where the node's final representation is placed at the root, and the nodes' representations that the parent nodes' representations computationally depend on are placed as its children. The computation graphs demonstrate which nodes are structurally consulted in forming a target node's representation, regardless of how their features are combined through an aggregation function. While the aggregation vary across models, our analysis is independent of them. What matters is the computation graph itself—i.e., which nodes appear in the aggregation paths and how frequently.

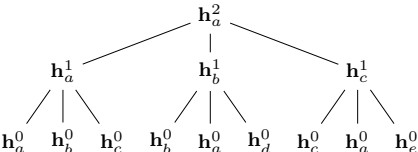

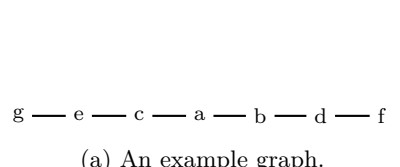

(a) An example graph.

(b) The computation graph of $\mathbf{h}_a^2$ for the graph in Figure 1a.

Figure 1: An example graph and the computation graph of a node in that graph produced by deterministic GGNNF.

The computation graph for $\mathbf{h}_a^2$ is shown in Figure 1b. This graph reveals that node $a$ (0-hop) is integrated 3 times, nodes $b$ and $c$ (1-hop) are integrated 2 times each, and nodes $d$ and $e$ (2-hop) are integrated once each. This uneven integration of neighbors at different hop distances can be measured using the average integration count (AIC), which quantifies on average how frequently nodes from each hop distance contribute to the final representation of a node. To formulate AIC, consider a node $i$ of node $v$'s $k$ hop distance is integrated $t_v^i$ times. The set of all nodes from node $v$'s $k$ hop distance is given by the hop function $H(k, v)$, where $H(k, v)$ contains all nodes whose shortest path to $v$ is exactly $k$, equivalently the set of nodes that can be reached in hops $k$ from node $v$. Therefore the average integration count of the $k$ hop nodes in $K$ layered GGNNF is $A_v^{k,K} = \frac{\sum\limits_{i \in H(k,v)} t_v^i}{|H(k,v)|}$. This metric provides a quantitative measure of the skewness in node integration across different hops. The higher the disparity in AIC values across hops, the greater the skewness, which can severely affect the GNN's ability to learn effective node representations. Table 1 shows how AIC values differ for node $a$ across different hops.

Next, to examine the effect of the sub-graph structure on AIC, we add a new node $h$ connected to nodes $a$ and $b$. The updated graph and the corresponding computation graph for $\mathbf{h}_a^2$ are shown in Figure 2a and Figure 2b, respectively. The updated computation graph shows that node $a$ (0-hop) is now integrated 4 times, while nodes $b$, $h$, and $c$ (1-hop) are integrated 3, 3, and 2 times, respectively, and nodes $d$ and $e$ (2-hop) are still integrated once each. The variation of AIC for the same node $a$ in graphs with different sub-graph structures indicates that AICs change as the sub-graph structure changes, as shown in Table 1.

Additionally, we analyze the effect of increasing the number of deterministic GGNNF layers on AIC. We extend the original two-layer GGNNF to three layers, producing the final representation $\mathbf{h}_a^3$ for the original

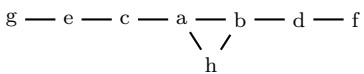

(a) Produced graph by adding another node to Figure 1a.

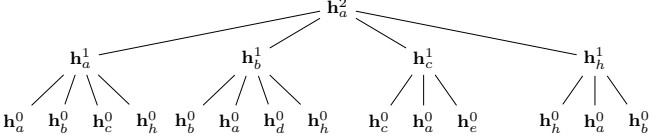

(b) The computation graph for $\mathbf{h}_a^2$ of the graph in the Figure 2a.

Figure 2: Produced graph by adding another node to Figure 1a and the computation graph of a node in that graph produced by deterministic GGNNF.

Table 1: Comparison of AIC values ($A_a^{k,K=2}$) for node $a$ with constant $K = 2$ and varying $k$, for the graphs of Figure 1a and Figure 2a, produced by the deterministic GGNNF.

| Different $k$ | $K = 2$ in original graph of Figure 1a | $K = 2$ in modified graph of Figure 2a |
|---|---|---|
| $k = 0$ | 3 | 4 |
| $k = 1$ | 2 | 2.67 |
| $k = 2$ | 1 | 1 |

graph in Figure 1a, whose computation graph is depicted in Figure 3. Here, node $a$ (0-hop) is integrated 7 times, nodes $b$ and $c$ (1-hop) are integrated 6 times each, nodes $d$ and $e$ (2-hop) are integrated 3 times each, and nodes $g$ and $f$ (3-hop) are integrated once each. The increased skewness is evident in the third column of Table 2, showing that adding more layers exacerbates the imbalance in AICs.

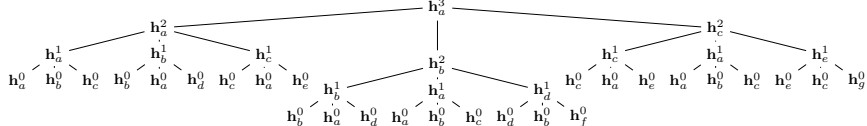

Figure 3: The computation graph for $\mathbf{h}_a^3$ of the graph in the Figure 1a produced by the deterministic GGNNF.

### 2.2.2 Randomized GGNNF

Instead of aggregating features from all neighboring nodes, GraphSAGE selects a random subset of neighbors through uniform sampling and aggregates features only from this subset. This modification replaces Line 4 of the Algorithm 1 with:

$$\mathbf{h}_{\mathcal{N}(v)}^k \leftarrow AGGREGATE_k(\{\mathbf{h}_u^{k-1}, \forall u \in \{u_1, u_2, u_3, \ldots, u_l \overset{i.i.d}{\sim} Uniform(\mathcal{N}(v))\}\})$$

Here, $l$ represents the size of the independent random sample taken from the neighborhood of node $v$. In this section, we analyze the implications of this sampling strategy on the node representation process.

To illustrate how randomized GGNNF generates the nodes' representation of a graph for GraphSAGE, let's consider the graph depicted in Figure 1a again. To capture interactions from nodes that are, for example, two hops away, we stack two layers of a randomized GGNNF, generating representations $\mathbf{h}_v^2$ for every node $v$. Also, consider that $l$, the independent random sample taken from the neighborhood of node $v$, is 1. Focusing on node **a**, its representation $\mathbf{h}_a^2$ is computed using its neighbor nodes' previous layer representation ($\mathbf{h}_b^1$ and $\mathbf{h}_c^1$) each with probability $\frac{1}{2}$, along with its own previous layer representation ($\mathbf{h}_a^1$) with probability 1. Node $b$'s representation $\mathbf{h}_b^1$, in turn, depends on its neighbor nodes' previous layer representation ($\mathbf{h}_a^0$ and $\mathbf{h}_d^0$) each with probability $\frac{1}{2}$ and its own previous layer representation ($\mathbf{h}_b^0$) with probability 1, and so on. We can visualize this iterative process through a randomized computation graph, where the node's final representation is placed at the root, and the nodes' representations that the parent nodes' representations computationally depend on are placed as its children, with corresponding probabilities.

The computation graph for $\mathbf{h}_a^2$ is shown in Figure 4. This graph reveals that node $a$ (0-hop) is integrated $\frac{3}{2}$ times (expected value), nodes $b$ and $c$ (1-hop) are integrated 1 times each (expected value), and nodes $d$ and $e$ (2-hop) are integrated $\frac{1}{4}$ times each (expected value). Table 3 shows how AIC values differ for node $a$ across different hops.

Table 2: Comparison of AIC values ($A_a^{k,K}$) for node $a$ with varying $k$ and $K$ on the same original graph of Figure 1a, produced by the deterministic GGNNF

| Different $k$ | $K = 2$ | $K = 3$ |
|---|---|---|
| $k = 0$ | 3 | 7 |
| $k = 1$ | 2 | 6 |
| $k = 2$ | 1 | 3 |
| $k = 3$ | - | 1 |

Figure 4: The computation graph of $\mathbf{h}_a^2$ for the graph in Figure 1a produced by the randomized GGNNF.

Next, to examine the effect of the sub-graph structure on AIC, we add a new node $h$ connected to nodes $a$ and $b$, as shown in Figure 2a. The corresponding computation graph for $\mathbf{h}_a^2$ is shown in Figure 5. The updated computation graph shows that node $a$ (0-hop) is now integrated $\frac{13}{9}$ times (expected value), while nodes $b$, $h$, and $c$ (1-hop) are integrated $\frac{5}{6}$, $\frac{7}{9}$, and $\frac{2}{3}$ times (expected value), respectively, and nodes $d$ and $e$ (2-hop) are integrated $\frac{1}{9}$ and $\frac{1}{6}$ times (expected value), respectively. The variation of AIC for the same node $a$ in graphs with different sub-graph structures indicates that AICs change as the sub-graph structure changes, as shown in Table 3.

Table 3: Comparison of AIC values ($A_a^{k,K=2}$) for node $a$ with constant $K = 2$ and varying $k$, for the graphs of Figure 1a and Figure 2a, produced by the randomized GGNNF

| Different $k$ | $K = 2$ | $K = 3$ |
|---|---|---|
| $k = 0$ | 1.5 | 1.44 |
| $k = 1$ | 1 | .76 |
| $k = 2$ | .25 | .14 |

Additionally, we analyze the effect of increasing the number of layers on AIC. We extend the original two-layer randomized GGNNF to three layers, producing the final representation $\mathbf{h}_a^3$ for the original graph in Figure 1a, whose computation graph is depicted in Figure 6. Here, node $a$ (0-hop) is integrated $\frac{5}{2}$ times (expected value), nodes $b$ and $c$ (1-hop) are integrated $\frac{15}{8}$ times each (expected value), nodes $d$ and $e$ (2-hop) are integrated $\frac{3}{4}$ times each (expected value), and nodes $g$ and $f$ (3-hop) are integrated $\frac{1}{8}$ times each (expected value). The increased skewness is evident in the third column of Table 4, showing that adding more layers exacerbates the imbalance in AICs.

As demonstrated in our examples, the K-skewed-traversal problem can arise even in simple topologies like straight-line graphs. Naturally, this issue persists—and often intensifies—in more complex graph structures due to their irregular neighborhood patterns. We include examples with more diversified and realistic graph structures in Appendix A to further illustrate the generality of the problem and strengthen our empirical justification.

Based on the analysis above on both deterministic and randomized GGNNF, we formulate the following propositions, which hold for any graphs for the GGNNF framework, rather than solely for the specific examples shown. These propositions illustrate how the K-skewed-traversal problem manifests in any graph structures:

**Proposition 1.** *GGNNFs prioritize certain hop neighbors over others when producing node representations.*

For a node, AICs of different hops, produced by a $K$ layered GGNNF are not necessarily the same, i.e.

$$\forall v \in \mathcal{V} : \forall k_1, k_2 \in [0, K] : (k_1 \neq k_2) \implies \neg\Box(A_v^{k_1,K} = A_v^{k_2,K})$$

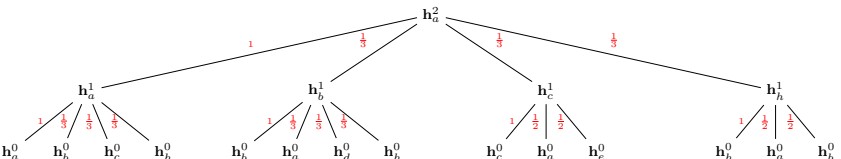

Figure 5: The computation graph for $\mathbf{h}_a^2$ of the graph in Figure 2a produced by randomized GGNNF, shows the effect of changing sub-graph structure on the average integration count.

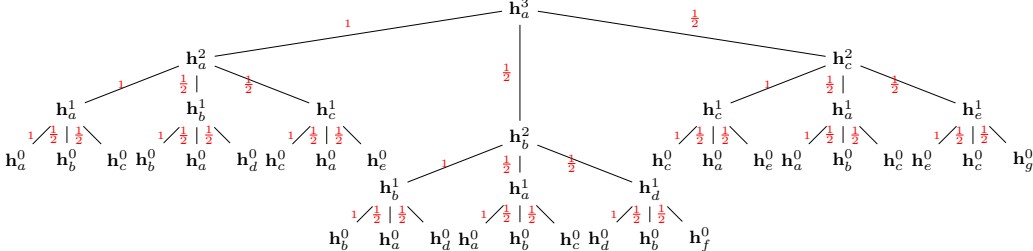

Figure 6: The computation graph for $\mathbf{h}_a^3$ of the graph in the Figure 1a produced by randomized GGNNF.

Consequently, when a node integrates information from nodes of multiple hops, GGNNFs give some hops higher priority than others.

**Proposition 2.** *This priority is influenced by the nodes' sub-graph structure.*

For two different nodes, AICs of the same hop, produced by a $K$ layered GGNNF are not necessarily the same, i.e.

$$\forall u, v \in \mathcal{V} : \forall k \in [0, K] : (u \neq v) \implies \neg\square(A_u^{k,K} = A_v^{k,K})$$

It means that the priorities depend on the nodes' sub-graph structure: a GGNNF may prioritize a node's first hop neighbors over the second hop's, whereas the same GGNNF may prioritize the opposite for another node.

**Proposition 3.** *Increasing the number of GGNNF layers exacerbates the skewness in prioritizing different hop neighbors.*

For a node, increasing the number of GGNNF layers does not reduce AIC for any hop, i.e.

$$\forall v \in \mathcal{V} : \forall k \in [0, min(K, L)] : (K > L) \implies (A_v^{k,K} \geq A_v^{k,L})$$

It means that the skewness among the priorities of a node's different hop neighbors does not decrease with the increase in the number of layers stacked in the GGNNF.

These propositions describe the fundamental properties of the K-skewed-traversal problem in any graphs which are not restricted to specific cases but are inherent to GGNNF models.

The objective of this paper is to develop a method that effectively addresses the K-skewed-traversal problem in GGNNF. To achieve this, we aim to ensure that each hop distance contributes uniformly to the node representation, maintaining an Average Integration Count (AIC) of 1 across all layers:

$$\forall v \in \mathcal{V}, \forall k \in [0, K] : A_v^{k,K} = 1$$

This uniform integration approach eliminates skewed prioritization across different hop distances. Additionally, we aim to adaptively adjust the importance of each hop during training, with the overall goal of enhancing representation learning and significantly improving performance, particularly in deep GNN architectures.

## 3 The Hop-wise Graph Attention Network (HGAT) Algorithm

Now, we introduce an algorithm that resolves the K-skewed-traversal problem in GGNNF. We introduce the Hop-wise Graph Attention Network (HGAT) Algorithm, which ensures uniform aggregation across different

Table 4: Comparison of AIC values ($A_a^{k,K}$) for node $a$ with varying $k$ and $K$ on the same original graph of Figure 1a, produced by the randomized GGNNF

| **Different** $k$ | $K = 2$ | $K = 3$ |
|---|---|---|
| $k = 0$ | 1.5 | 2.5 |
| $k = 1$ | 1 | 1.875 |
| $k = 2$ | .25 | .75 |
| $k = 3$ | - | .125 |

hop distances (See Theorem 1) and incorporates a learnable hop-wise attention mechanism. This approach allows the algorithm to adaptively attend to specific hop distances, improving node representations for downstream tasks such as node classification or link prediction.

### 3.1 Algorithm Description

In GGNNF, nodes aggregate information from their neighbors across multiple hops. However, this process often results in a skewed AIC, especially in deep GNN architectures. This skewed aggregation can degrade the quality of the node representations, leading to poor results.

To address this, HGAT performs two core components: Uniform Aggregation Across Hops (Phase 1) followed by Hop-wise Attention Mechanism (Phase 2).

**Uniform Aggregation Across Hops (Phase 1):** The core idea is to treat each hop separately during aggregation. To create a node's representation, we take its hop neighbors from 0 up to $K$ separately and compute hop-specific summaries (each hop gets its own summary) by aggregating the features of nodes at that hop distance. This step prevents the skewed aggregation seen in GGNNF, ensuring each hop's information is integrated equally.

**Hop-wise Attention Mechanism (Phase 2):** The hop-wise summaries are then combined using attention weights learned during training. This mechanism allows the network to prioritize specific hop distances, resulting in better node representations.

---

**Algorithm 2** Representation generation of HGAT

---

**Require:** Graph $\mathcal{G}(\mathcal{V}, \mathcal{E})$; input features $\{\mathbf{x}_v, \forall v \in \mathcal{V}\}$; depth $K$; weight matrices $\mathbf{W}^k$, $\forall k \in \{0, \ldots, K\}$; non-linearity $\sigma$; differentiable aggregator functions $AGGREGATE_k, \forall k \in \{1, \ldots, K\}$; hop function $H : (k, v) \rightarrow$ set of node $v$'s $k$'th hop neighborhood

**Ensure:** Vector representations $\mathbf{z}_v$ for all $v \in \mathcal{V}$

1: $\mathbf{h}_v^0 \leftarrow \mathbf{x}_v, \forall v \in \mathcal{V}$;
2: **for** $v \in \mathcal{V}$ **do**
3:      **for** $k = 0 \ldots K$ **do**
4:          $\mathbf{c}_v^k \leftarrow AGGREGATE(\mathbf{x}_v, \forall v \in H(k, v))$        ▷ Mean or sum aggregation
5:          $\mathbf{n}_v^k \leftarrow \sigma(\mathbf{W}^k \cdot \mathbf{c}_v^k)$        ▷ Compute hop-wise summary
6:      **end for**
7:      **for** $k = 0 \ldots K$ **do**
8:          $\lambda_k \leftarrow \frac{e^{\lambda_k}}{\sum_{i=0}^{K} e^{\lambda_i}}$        ▷ Learn attention weights
9:      **end for**
10:      $\mathbf{h}_v^K \leftarrow \sum_{k=0}^{K} \lambda_k \mathbf{n}_v^k$
11: **end for**
12: $\mathbf{z}_v \leftarrow \mathbf{h}_v^K, \forall v \in \mathcal{V}$        ▷ Compute final node representation

---

Algorithm 2 describes the forward propagation process in HGAT, which generates node representations while resolving the K-skewed-traversal problem. Algorithm 2 works as follows. For every node $v$ in the graph $\mathcal{G}(\mathcal{V}, \mathcal{E})$, the initial representation $\mathbf{h}_v^0$ is set to its input features $\mathbf{x}_v$ in Line 1, as a preparation for Uniform Aggregation Across Hops (Phase 1). The **for** loop of Lines 2-11 iterates over every node $v$. Within

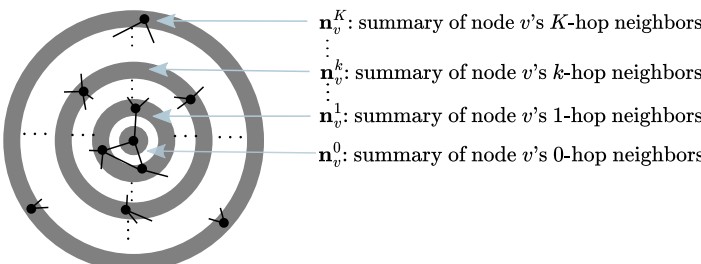

$\mathbf{n}_v^K$: summary of node $v$'s $K$-hop neighbors

$\mathbf{n}_v^k$: summary of node $v$'s $k$-hop neighbors

$\mathbf{n}_v^1$: summary of node $v$'s 1-hop neighbors

$\mathbf{n}_v^0$: summary of node $v$'s 0-hop neighbors

Figure 7: HGAT creates separate summaries of a node's every neighboring hops, including the node itself, and integrates these summaries to generate node representations.

this, the **for** loop in Lines 3-6 creates $K + 1$ separate hop-specific summaries for every node. The $k$'th summary of node $v$ is created by

(i) Aggregating the features of all nodes in the $k$-hop neighborhood of $v$ in Line 4:

$$\mathbf{c}_v^k \leftarrow AGGREGATE(\mathbf{x}_v, \forall v \in H(k, v))$$

(ii) Transforming it using a weight matrix $\mathbf{W}^k$ followed by a non-linear activation function $\sigma$ in Line 5:

$$\mathbf{n}_v^k \leftarrow \sigma(\mathbf{W}^k \cdot \mathbf{c}_v^k)$$

The creation of the separate hop-specific summaries is depicted in Figure 7. The aggregation function in Line 4 can be any differentiable operation, such as mean or sum. The version of HGAT that uses mean aggregation is referred to as HGAT-mean, while the version that employs sum aggregation is called HGAT-sum.

To generate hop-specific summaries,$k$-hop neighborhoods are computed using breadth-first search (BFS) (or depth-limited DFS) up to depth $k$. The time complexity for computing the $k$-hop neighborhood of a node is approximately $\frac{D^{k+1}-1}{D-1}$ where $D$ is the average node degree. This process is easily parallelizable, since the $k$-hop neighborhood of each node can be computed independently of others. In practice, we precompute these neighborhoods before model training, ensuring scalability for moderately sized graphs. The creation of the hop-wise summaries concludes Uniform Aggregation Across Hops (Phase 1).

Once the hop-wise summaries $\mathbf{n}_v^k$ are computed for each hop, the algorithm proceeds to the Hop-wise Attention Mechanism (Phase 2). In this phase, the attention weights $\lambda_k$ determine the contribution of $\mathbf{n}_v^k$ to the final node representation. These weights are learned during training and Line 8 constrains them to sum to 1 using the softmax function, ensuring balanced importance across different hops and preventing extreme values.

$$\lambda_k \leftarrow \frac{e^{\lambda_k}}{\sum_{i=0}^{K} e^{\lambda_i}}$$

Line 10 computes the final node representation as a weighted sum of the hop-wise summaries. The weights, determined by the attention mechanism, control the contribution of each hop-wise summary—higher attention weights result in a greater influence of the corresponding summary on the weighted sum. The weighted sum of the hop-wise summaries is computed as:

$$\mathbf{h}_v^K \leftarrow \sum_{k=0}^{K} \lambda_k \mathbf{n}_v^k$$

Overall, HGAT explicitly addresses the limitations of existing GGNNF architectures by eliminating the skew in hop contributions and allowing the model to learn which hops are most important. By treating each hop separately and applying a learnable attention mechanism, HGAT ensures that all neighborhood information is utilized effectively, regardless of the hop distance. This not only results in more balanced node integration but also allows the algorithm to adaptively focus on the most relevant features during training. As a consequence, HGAT enhances the expressive power of GNNs, particularly in deep architectures. The

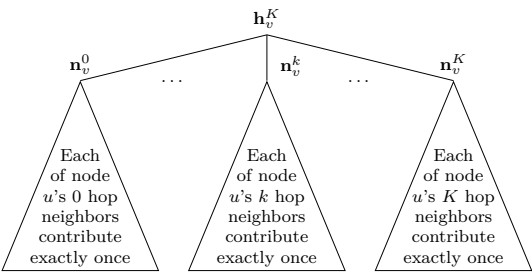

Figure 8: The computation graph produced during Uniform Aggregation Across Hops (Phase 1) of HGAT.

next section demonstrates how Uniform Aggregation Across Hops (Phase 1) mitigates the K-skewed-traversal problem by ensuring equal contribution from each hop, thereby establishing a foundation for the Hop-wise Attention Mechanism (Phase 2).

## 4 Theoretical Analysis

**Theorem 1.** *HGAT ensures AIC value of 1 in Uniform Aggregation Across Hops (Phase 1) before it applies Hop-wise Attention Mechanism (Phase 2).*

$$\forall v \in \mathcal{V}, \forall k \in [0, K] : A_v^{k,K} = 1$$

*Proof.* Let's consider a node $v$ for which we are generating a representation integrating information up to $K$-hops, meaning we are interested in calculating $\mathbf{h}_v^K$, the final representation of node $v$. This representation $\mathbf{h}_v^K$ is computed from the hop-wise summaries $\mathbf{n}_v^0, \mathbf{n}_v^1, \ldots, \mathbf{n}_v^k, \ldots, \mathbf{n}_v^{K-1}, \mathbf{n}_v^K$, where $\mathbf{n}_v^k$ represents the summary of nodes at the $k$-th hop. The computation graph is shown in Figure 8.

AIC of node $v$ for the nodes in its $k$-th hop for a $K$ layer HGAT is $A_v^{k,K}$. A node $i$ of node $v$'s $k$ hop distance is integrated $t_v^i$ times. The set of all nodes from node $v$'s $k$ hop distance is given by the hop function $H(k, v)$. To compute the hop-wise summary $\mathbf{n}_v^k$, we aggregate the features of all nodes at the $k$-hop distance from node $v$.

AIC $A_v^{k,K}$ is calculated as:

$$A_v^{k,K} = \frac{\sum\limits_{i \in H(k,v)} t_v^i}{|H(k,v)|}$$

The hop-wise summary, $\mathbf{n}_v^k$ is calculated by taking the summation or average of $v$'s $k$-th hop neighbors, unlike GGNNF where each neighboring node can contribute multiple times randomly (See Section 2 and Section 3 for detailed analysis). Consequently, each node $i$ within $k$-th hop neighborhood contributes to the summary exactly once in Phase 1, meaning $t_v^i = 1$. Thus, the sum simplifies to:

$$A_v^{k,K} = \frac{\sum\limits_{i \in H(k,v)} 1}{|H(k,v)|} = \frac{|H(k,v)|}{|H(k,v)|} = 1$$

$\square$

This theorem demonstrates that AIC for each hop in Uniform Aggregation Across Hops (Phase 1) is 1. Consequently, this phase effectively mitigates the K-skewed-traversal problem inherent in GGNNF models by ensuring that each hop distance contributes uniformly before applying the Hop-wise Attention Mechanism (Phase 2). This prevents HGAT's attention weights from being applied to already skewed node representations.

## 5 Experimental Evaluation

This section evaluates the proposed Hop-wise Graph Attention Network (HGAT) on standard node classification tasks, comparing its performance with GCN, GraphSAGE, and GAT (the detailed reason for selecting

these datasets is discussed in Section 2). The primary objectives of the experiments are to (i) demonstrate the effectiveness of the hop-wise attention mechanism, and (ii) validate the method's ability to mitigate the K-skewed-traversal problem.

We conduct experiments on three widely-used benchmark citation network datasets: Cora (Yang et al., 2016), Citeseer (Yang et al., 2016), and Pubmed (Yang et al., 2016) provided by PyTorch Geometric framework (Fey & Lenssen, 2019). These datasets consist of nodes representing documents and edges denoting citations between them. The features for each node are bag-of-words representations of the documents, and the task is to classify each node into one of several predefined classes. Table 5 provides key statistics of these datasets.

Table 5: Statistics of the dataset used in our experiment

| Dataset | Nodes | Edges | Features | Classes |
|---------|-------|-------|----------|---------|
| Cora | 2,708 | 10,556 | 1,433 | 7 |
| Citeseer | 3,327 | 9,104 | 3,703 | 6 |
| Pubmed | 19,717 | 88,648 | 500 | 3 |

### 5.1 Impact of the K-skewed-traversal Problem on the Datasets

To assess how the working procedure of the GGNNF introduces skewness in these datasets, we measure AIC of all nodes in each dataset. Since it is impractical to inspect AICs for all nodes simultaneously, we summarize the results by calculating the mean ($\mu_{A_v^{k,K}} = \frac{\sum_{u \in V} A_u^{k,K}}{|V|}$) and standard deviation ($\sigma_{A_v^{k,K}} = \sqrt{\frac{\sum_{u \in V} (A_u^{k,K} - \mu_{A_v^{k,K}})^2}{|V|}}$) of the integration counts, where $A_v^{k,K}$ represents AIC of node $v$ at hop $k$ with $K$ layers GNN.

Table 6 shows the mean and standard deviation of AICs on varying hops $k$ with static GNN layers ($K = 4$) to see the effect of the number of hops on AICs. The varying mean of the AIC illustrates that AICs of different hops are not necessarily the same, the effect stated in the Proposition 1. Moreover, the standard deviation of a specific hop $k$ shows that AIC varies from node to node with the same hop $k$ and the same GGNNF layers $K$, as the sub-graph structure of the node of interest changes, the effect stated in the Proposition 2.

Additionally, Table 7 shows the mean and standard deviation of AICs on varying GGNNF layers ($K$) with static hop $k = 2$. The increasing AIC demonstrates that increasing the GGNNF layer will increase AIC of a node's specific hop, the effect stated in the Proposition 3.

### 5.2 Experimental Results

We now present our comparative empirical evaluation on the Cora, Citeseer, and Pubmed datasets. We split the datasets into three categories. The training set is created by randomly selecting 20 nodes per class. The remaining nodes are randomly divided into a validation set containing 500 nodes and a test set with the remaining nodes. We optimize hyperparameters separately for each method, train for 600 epochs at every K, and report the best test accuracy observed across all epochs. All experiments are carried out on a machine equipped with an NVIDIA GeForce RTX 3090 GPU (24 GB of RAM), an AMD Ryzen 9 5950X 16-Core (4.9 GHz processor), and 98GB RAM.

The results of our comparative empirical evaluation on the Cora, Citeseer, and Pubmed datasets are summarized in Figures 9a, 9b, and 9c, respectively. These figures illustrate how the node classification accuracy of each method changes as the number of GNN layers increases. To complement these plots with precise numerical values, detailed accuracy results for each dataset and method across all depth settings are provided in Tables 8, 9, and 10 in Appendix B.

Across all datasets, we observe a general trend: while all methods perform similarly with a lower number of GNN layers, the baseline methods (GCN, GraphSAGE-mean, GraphSAGE-sum, and GAT) experience significant drops in accuracy as the number of layers increases. In contrast, our proposed method, HGAT-mean and HGAT-sum, shows a much more gradual decrease in accuracy. For example, on the Cora dataset (Figure 9a), GCN's accuracy drops by approximately 50% on average after 15 layers, while HGAT-mean

Table 6: Impact of varying hops ($k$) on the mean and standard deviation of the AICs (with static GGNNF layers $K = 4$)

| | Cora | | Citeseer | | Pubmed | |
|---|---|---|---|---|---|---|
| | $\mu_{A_v^{k,K}}$ | $\sigma_{A_v^{k,K}}$ | $\mu_{A_v^{k,K}}$ | $\sigma_{A_v^{k,K}}$ | $\mu_{A_v^{k,K}}$ | $\sigma_{A_v^{k,K}}$ |
| $k = 0$ | 133.77 | 692.13 | 75.51 | 318.60 | 255.93 | 949.40 |
| $k = 1$ | 119.76 | 142.88 | 63.99 | 123.95 | 177.46 | 264.00 |
| $k = 2$ | 58.68 | 52.51 | 33.46 | 52.59 | 88.62 | 114.61 |
| $k = 3$ | 9.82 | 6.13 | 7.28 | 9.43 | 15.51 | 16.29 |
| $k = 4$ | 1.77 | 0.95 | 1.31 | 1.33 | 2.58 | 1.57 |

Table 7: Impact of varying GGNNF layers ($K$) on the mean and standard deviation of the AICs (with static layer $k = 2$)

| | Cora | | Citeseer | | Pubmed | |
|---|---|---|---|---|---|---|
| | $\mu_{A_v^{k,K}}$ | $\sigma_{A_v^{k,K}}$ | $\mu_{A_v^{k,K}}$ | $\sigma_{A_v^{k,K}}$ | $\mu_{A_v^{k,K}}$ | $\sigma_{A_v^{k,K}}$ |
| $K = 2$ | 1.08 | 0.34 | 0.92 | 0.52 | 1.09 | 0.18 |
| $K = 3$ | 5.19 | 2.66 | 3.91 | 3.23 | 5.07 | 3.86 |
| $K = 4$ | 58.68 | 52.51 | 33.46 | 52.59 | 88.62 | 114.61 |

maintains a much smaller 35% average drop and HGAT-sum always maintains steady accuracy. This is because HGAT maintains uniform contribution from each hop, solving the K-skewed-traversal problem that other methods fail to address. Figure 9b shows that in the Citeseer dataset, baseline methods experience steep declines in accuracy after around 12 layers, while HGAT-mean maintains a steady and gradual decrease but higher than other baseline methods, preserving higher classification performance. HGAT-sum maintains higher accuracy than the baseline methods but lower than the HGAT-mean, preserving steady accuracy. Here, the reason for HGAT's higher accuracy as the number of layers increases is stated above. As seen in Figure 9c, in the Pubmed dataset, HGAT-mean maintains a higher accuracy in contrast to baseline methods, whose performance drastically deteriorates after 10 layers, while HGAT-sum maintains steady accuracy even after around 10 layers, for the same reasons mentioned.

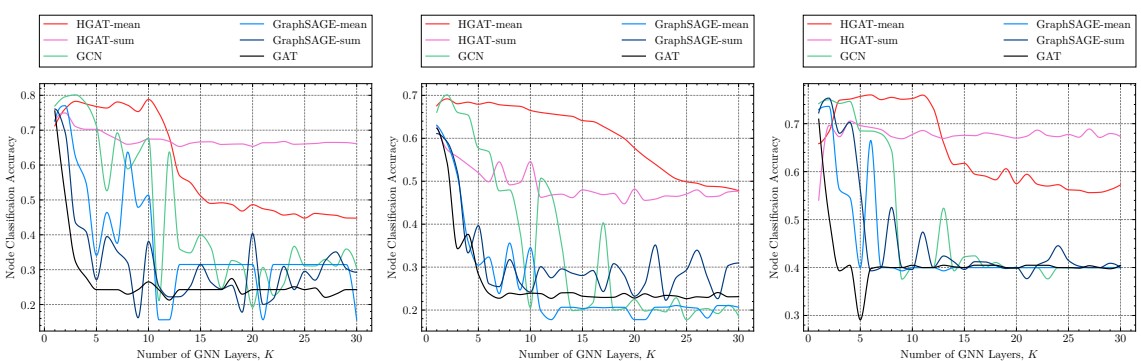

(a) Node classification accuracy for different methods in Cora.
(b) Node classification accuracy for different methods in Citeseer.
(c) Node classification accuracy for different methods in Pubmed.

Figure 9: Node classification accuracy for different methods in Cora, Citeseer, and Pubmed datasets.

Notably, HGAT performs substantially better than the state-of-the-art methods, its slight decrease in accuracy as the number of layers increases is due to the well-known issues in deep GGNNF architectures: over-smoothing and over-squashing, which are also present in the state-of-the-art methods. However, the decline in HGAT is far less pronounced than that observed in the state-of-the-art methods.

In contrast, the significant performance degradation in the state-of-the-art methods is further exacerbated by the proposed K-skewed-traversal problem, which we demonstrate arises in deep GGNNF architectures. This problem intensifies the decline by prioritizing different hop neighbors unevenly, leading to inefficient learning from multi-hop nodes.

On the other hand, HGAT effectively addresses the K-skewed-traversal problem by ensuring uniform aggregation across hops and incorporating a hop-wise attention mechanism that balances the contributions of neighbors at various hop distances. However, all three issues—over-smoothing, over-squashing, and the K-skewed-traversal problem remain prevalent in the state-of-the-art methods.

These results demonstrate HGAT's robustness in mitigating the K-skewed-traversal problem that arises from deep GGNNF architectures. By maintaining a more balanced integration of information from different hop distances, HGAT achieves superior performance on node classification tasks across multiple datasets.

## 6 Conclusions and Future Work

In this work, we identify a new problem inherent in traditional Graph Neural Networks (GNNs) named the K-skewed-traversal problem and introduce the Hop-wise Graph Attention Network (HGAT) algorithm to address it. Through our experiments on benchmark datasets, HGAT demonstrates significant improvements in node classification accuracy in deep GNN architectures, by mitigating the skewed priority given to different hop neighbors. The introduction of the hop-wise attention mechanism ensures an adaptive integration of information from various hop distances, allowing HGAT to maintain high performance as the network depth increases. These results highlight the potential of HGAT to enhance the effectiveness of GNNs in real-world applications, particularly in scenarios requiring deep architectures. In the future, we would like to explore the adaptation of HGAT to other tasks within graph learning such as edge prediction, graph classification, etc. We also want to explore whether HGAT can be extended for larger and more complex graph structures e.g. heterogeneous graphs. In general, the advances presented in this paper contribute to the ongoing development of robust and scalable GNN models, paving the way for their broader application across various fields.

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

## A  Graph Examples Illustrating the K-skewed-traversal Problem

In addition to Figures 1a and 2a, we present a range of additional graph examples to further illustrate the K-skewed-traversal problem. When generating a node's representation, GGNNF processes a subgraph centered on the node, with radius defined by the depth $K$. These $K$-radius subgraphs can vary significantly in structure and shape depending on the target node's location and neighborhood within the overall graph. Figure 10 illustrates how GGNNF computes each node's representation from its surrounding sub-graph (centered at the target node) up to radius $K$. This results in different sub-graphs for different nodes, depending on their neighborhoods.

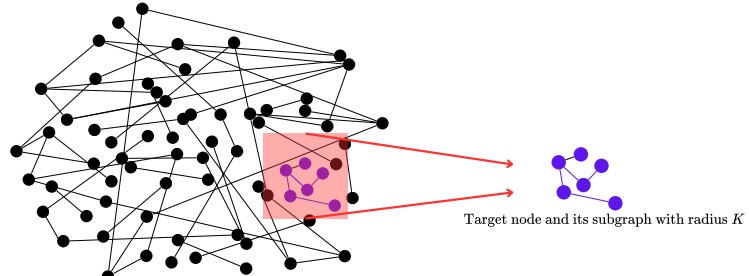
Target node and its subgraph with radius $K$

Figure 10: GGNNF computes each node's representation using its surrounding sub-graph, centered on the target node and extending up to radius $K$. Thus it encounters different sub-graph for different nodes.

Figure 11 illustrates, for deterministic GGNNF, how nodes at different hop distances contribute to a target node's representation, making the skew visually apparent. The same type of analysis and visualization applies to randomized GGNNF as well. The figure compares node contributions for $K = 2$ (left) and $K = 3$ (right) using the same sub-graph. Across all examples, we observe uneven contributions from different hop distances, confirming Proposition 1. Increasing $K$ increases contributions from multiple hops, confirming Proposition 3. Additionally, the contribution of each node varies with changes in the subgraph structure, aligning with the claim in Proposition 2. These examples demonstrate that the K-skewed-traversal problem occurs across a variety of graph topologies, and that its severity can change with both the number of layers $K$ and the node's local sub-graph structure.

## B  Detailed Results

To complement the plots in Figures 9a, 9b, and 9c, we provide full numerical accuracy values for each dataset and method across different layer depths ($K$) in Tables 8, 9, 10.

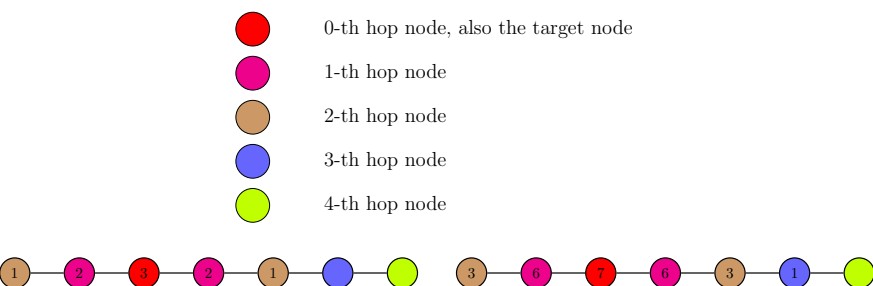

(a) Both examples show the disparity of contribution of different nodes validating Proposition 1. Additionally, comparing left one with right one shows contribution is increased as the number of layers increased from $K = 2$ to $K = 3$, validating Proposition 3.

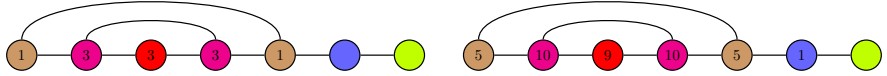

(b) Both examples show the disparity of contribution of different nodes validating Proposition 1. Additionally, comparing left one with right one shows contribution is increased as the number of layers increased from $K = 2$ to $K = 3$, validating Proposition 3. Furthermore, in $K = 3$, hop 1 is contributing more than hop 0, which is different from previous three graphs.

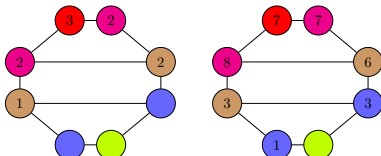

(c) Both examples show the disparity of contribution of different nodes validating Proposition 1. Additionally, comparing left one with right one shows contribution is increased as the number of layers increased from $K = 2$ to $K = 3$, validating Proposition 3. Furthermore, in $K = 3$, hop 1 is contributing more than hop 0, which is different from previous three graphs.

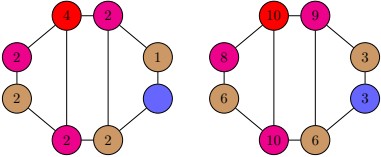

(d) Both examples show the disparity of contribution of different nodes validating Proposition 1. Additionally, comparing left one with right one shows contribution is increased as the number of layers increased from $K = 2$ to $K = 3$, validating Proposition 3.

Figure 11: Left one showing individual node's contribution on generating target node's representation for $K = 2$, right one for $K = 3$ for the same sub-graph. These comparison shows the validity of K-skewed-traversal.

Table 8: Node classification accuracy (%) of HGAT-mean, HGAT-sum, GCN, GraphSAGE-mean, GraphSAGE-sum and GAT on the Cora dataset across different numbers of layers ($K$). *Note: Bold values indicate the best performance at each depth ($K$).*

| Layers | HGAT-mean | HGAT-sum | GCN | GraphSAGE-mean | GraphSAGE-sum | GAT |
|---|---|---|---|---|---|---|
| 0 | 0.545 | 0.517 | **0.770** | 0.727 | 0.758 | 0.748 |
| 1 | 0.712 | 0.728 | **0.769** | 0.725 | 0.761 | 0.755 |
| 2 | 0.762 | 0.750 | **0.796** | 0.770 | 0.693 | 0.512 |
| 3 | 0.783 | 0.710 | **0.801** | 0.622 | 0.434 | 0.322 |
| 4 | 0.775 | 0.703 | **0.777** | 0.549 | 0.409 | 0.278 |
| 5 | **0.768** | 0.703 | 0.712 | 0.341 | 0.271 | 0.243 |
| 6 | **0.764** | 0.689 | 0.526 | 0.464 | 0.395 | 0.243 |
| 7 | **0.781** | 0.674 | 0.692 | 0.375 | 0.353 | 0.243 |
| 8 | **0.772** | 0.660 | 0.589 | 0.637 | 0.320 | 0.230 |
| 9 | **0.753** | 0.664 | 0.632 | 0.479 | 0.162 | 0.243 |
| 10 | **0.788** | 0.675 | 0.677 | 0.514 | 0.381 | 0.265 |
| 11 | **0.750** | 0.675 | 0.211 | 0.156 | 0.252 | 0.243 |
| 12 | **0.679** | 0.668 | 0.638 | 0.156 | 0.222 | 0.213 |
| 13 | 0.569 | **0.653** | 0.359 | 0.315 | 0.222 | 0.243 |
| 14 | 0.550 | **0.662** | 0.348 | 0.315 | 0.251 | 0.243 |
| 15 | 0.512 | **0.666** | 0.399 | 0.315 | 0.315 | 0.243 |
| 16 | 0.490 | **0.667** | 0.363 | 0.315 | 0.264 | 0.243 |
| 17 | 0.492 | **0.659** | 0.246 | 0.315 | 0.245 | 0.243 |
| 18 | 0.487 | **0.660** | 0.326 | 0.315 | 0.255 | 0.275 |
| 19 | 0.468 | **0.661** | 0.314 | 0.315 | 0.178 | 0.230 |
| 20 | 0.486 | **0.654** | 0.192 | 0.315 | 0.404 | 0.243 |
| 21 | 0.475 | **0.664** | 0.307 | 0.156 | 0.200 | 0.243 |
| 22 | 0.469 | **0.664** | 0.226 | 0.315 | 0.217 | 0.243 |
| 23 | 0.456 | **0.668** | 0.259 | 0.315 | 0.309 | 0.243 |
| 24 | 0.460 | **0.660** | 0.367 | 0.315 | 0.243 | 0.251 |
| 25 | 0.448 | **0.661** | 0.311 | 0.315 | 0.295 | 0.243 |
| 26 | 0.461 | **0.662** | 0.310 | 0.315 | 0.270 | 0.248 |
| 27 | 0.458 | **0.665** | 0.330 | 0.315 | 0.326 | 0.221 |
| 28 | 0.456 | **0.665** | 0.310 | 0.315 | 0.351 | 0.230 |
| 29 | 0.448 | **0.664** | 0.359 | 0.315 | 0.303 | 0.243 |
| 30 | 0.448 | **0.661** | 0.304 | 0.156 | 0.293 | 0.243 |

Table 9: Node classification accuracy (%) of HGAT-mean, HGAT-sum, GCN, GraphSAGE-mean, GraphSAGE-sum and GAT on the Citeseer dataset across different numbers of layers ($K$). *Note: Bold values indicate the best performance at each depth ($K$).*

| Layers | HGAT-mean | HGAT-sum | GCN | GraphSAGE-mean | GraphSAGE-sum | GAT |
|--------|-----------|----------|-------|----------------|---------------|-------|
| 0 | 0.586 | 0.587 | **0.661** | 0.631 | 0.614 | 0.638 |
| 1 | **0.676** | 0.629 | 0.661 | 0.631 | 0.611 | 0.624 |
| 2 | 0.692 | 0.574 | **0.701** | 0.588 | 0.592 | 0.543 |
| 3 | **0.680** | 0.557 | 0.661 | 0.511 | 0.520 | 0.343 |
| 4 | **0.684** | 0.539 | 0.654 | 0.361 | 0.333 | 0.376 |
| 5 | **0.679** | 0.520 | 0.577 | 0.305 | 0.397 | 0.283 |
| 6 | **0.684** | 0.499 | 0.569 | 0.324 | 0.262 | 0.239 |
| 7 | **0.678** | 0.545 | 0.478 | 0.238 | 0.255 | 0.228 |
| 8 | **0.676** | 0.492 | 0.480 | 0.356 | 0.318 | 0.239 |
| 9 | **0.675** | 0.497 | 0.383 | 0.247 | 0.263 | 0.236 |
| 10 | **0.665** | 0.546 | 0.205 | 0.345 | 0.242 | 0.239 |
| 11 | **0.660** | 0.463 | 0.507 | 0.198 | 0.301 | 0.238 |
| 12 | **0.657** | 0.468 | 0.471 | 0.178 | 0.275 | 0.227 |
| 13 | **0.654** | 0.470 | 0.352 | 0.207 | 0.296 | 0.240 |
| 14 | **0.651** | 0.462 | 0.199 | 0.207 | 0.286 | 0.240 |
| 15 | **0.641** | 0.480 | 0.201 | 0.204 | 0.280 | 0.232 |
| 16 | **0.639** | 0.475 | 0.217 | 0.207 | 0.292 | 0.231 |
| 17 | **0.628** | 0.469 | 0.403 | 0.205 | 0.243 | 0.229 |
| 18 | **0.614** | 0.472 | 0.200 | 0.207 | 0.308 | 0.230 |
| 19 | **0.600** | 0.447 | 0.204 | 0.207 | 0.281 | 0.239 |
| 20 | **0.577** | 0.482 | 0.225 | 0.178 | 0.232 | 0.228 |
| 21 | **0.557** | 0.455 | 0.198 | 0.178 | 0.261 | 0.238 |
| 22 | **0.540** | 0.458 | 0.201 | 0.207 | 0.352 | 0.230 |
| 23 | **0.521** | 0.466 | 0.196 | 0.207 | 0.223 | 0.234 |
| 24 | **0.506** | 0.465 | 0.229 | 0.211 | 0.276 | 0.233 |
| 25 | **0.499** | 0.470 | 0.177 | 0.207 | 0.294 | 0.226 |
| 26 | **0.495** | 0.480 | 0.199 | 0.205 | 0.339 | 0.231 |
| 27 | **0.488** | 0.464 | 0.204 | 0.181 | 0.277 | 0.229 |
| 28 | **0.488** | 0.465 | 0.192 | 0.211 | 0.226 | 0.241 |
| 29 | **0.484** | 0.475 | 0.211 | 0.211 | 0.304 | 0.231 |
| 30 | **0.478** | 0.477 | 0.185 | 0.207 | 0.310 | 0.231 |

Table 10: Node classification accuracy (%) of HGAT-mean, HGAT-sum, GCN, GraphSAGE-mean, GraphSAGE-sum and GAT on the Pubmed dataset across different numbers of layers ($K$). *Note: Bold values indicate the best performance at each depth ($K$).*

| Layers | HGAT-mean | HGAT-sum | GCN | GraphSAGE-mean | GraphSAGE-sum | GAT |
|---|---|---|---|---|---|---|
| 0 | 0.456 | 0.426 | **0.742** | 0.729 | 0.722 | 0.717 |
| 1 | 0.658 | 0.540 | **0.742** | 0.729 | 0.722 | 0.709 |
| 2 | 0.684 | 0.696 | 0.750 | 0.736 | **0.753** | 0.505 |
| 3 | **0.749** | 0.673 | 0.742 | 0.562 | 0.680 | 0.393 |
| 4 | **0.751** | 0.706 | 0.747 | 0.546 | 0.703 | 0.405 |
| 5 | **0.757** | 0.696 | 0.685 | 0.400 | 0.562 | 0.291 |
| 6 | **0.760** | 0.692 | 0.685 | 0.665 | 0.393 | 0.397 |
| 7 | **0.750** | 0.688 | 0.679 | 0.400 | 0.400 | 0.401 |
| 8 | **0.755** | 0.673 | 0.644 | 0.400 | 0.525 | 0.400 |
| 9 | **0.751** | 0.669 | 0.376 | 0.393 | 0.401 | 0.400 |
| 10 | **0.753** | 0.677 | 0.400 | 0.400 | 0.396 | 0.405 |
| 11 | **0.760** | 0.686 | 0.400 | 0.400 | 0.474 | 0.400 |
| 12 | **0.733** | 0.675 | 0.400 | 0.400 | 0.400 | 0.400 |
| 13 | 0.660 | **0.669** | 0.524 | 0.393 | 0.424 | 0.405 |
| 14 | 0.615 | **0.674** | 0.393 | 0.400 | 0.412 | 0.402 |
| 15 | 0.617 | **0.676** | 0.422 | 0.400 | 0.396 | 0.400 |
| 16 | 0.595 | **0.675** | 0.424 | 0.400 | 0.412 | 0.405 |
| 17 | 0.591 | **0.681** | 0.400 | 0.400 | 0.412 | 0.405 |
| 18 | 0.583 | **0.678** | 0.411 | 0.400 | 0.404 | 0.405 |
| 19 | 0.606 | **0.674** | 0.400 | 0.400 | 0.398 | 0.400 |
| 20 | 0.575 | **0.670** | 0.400 | 0.400 | 0.400 | 0.399 |
| 21 | 0.595 | **0.674** | 0.400 | 0.400 | 0.377 | 0.405 |
| 22 | 0.574 | **0.687** | 0.400 | 0.400 | 0.405 | 0.405 |
| 23 | 0.570 | **0.675** | 0.376 | 0.400 | 0.415 | 0.405 |
| 24 | 0.573 | **0.673** | 0.400 | 0.400 | 0.445 | 0.400 |
| 25 | 0.562 | **0.678** | 0.400 | 0.400 | 0.415 | 0.400 |
| 26 | 0.561 | **0.672** | 0.400 | 0.400 | 0.403 | 0.399 |
| 27 | 0.556 | **0.689** | 0.400 | 0.400 | 0.400 | 0.404 |
| 28 | 0.557 | **0.671** | 0.400 | 0.400 | 0.400 | 0.400 |
| 29 | 0.562 | **0.679** | 0.400 | 0.400 | 0.409 | 0.397 |
| 30 | 0.572 | **0.673** | 0.400 | 0.400 | 0.400 | 0.405 |

