# OpenReview forum: "Addressing Node Integration Skewness in Graph Neural Networks Using Hop-Wise Attention"
_TMLR — Rejected by TMLR_

### Review · Reviewer_z97w · 2025-06-05

**Summary Of Contributions:**

This paper proposes a novel GNN model that addresses the K-skewed-traversal problem in graph neural networks.

**Audience:**

No

**Claims And Evidence:**

Yes

**Requested Changes:**

See Weaknesses and Suggestions.

**Strengths And Weaknesses:**

Strengths:

1. The paper presents the HGAT algorithm with clear writing.
2. The figures in the algorithm section illustrate the novelty effectively.
3. The dataset tables are clean and well-organized.

Weaknesses and Suggestions:

1.  The paper lacks numerical performance results when comparing HGAT to baseline models.  Figure 9 only provides a plot-based comparison. You need to include at least one table that presents numerical results to allow readers to quantitatively assess the improvements of HGAT.
2.  If I understand Figure 9 correctly, For the Cora dataset, all methods appear to reach their highest scores at layers 3 or 4. For the Citeseer dataset, the peak occurs between layers 2 and 4. However, it seems that in both datasets, GCN achieves the highest absolute performance, not HGAT. It currently seems inconsistent with the claim of HGAT’s superiority.
3.  Followed by point 2, while HGAT shows an advantage in handling deeper layers, the practical usefulness of deeper layers in these specific datasets is questionable, as performance generally degrades with increased depth across all methods. To better demonstrate the benefits of HGAT in deeper architectures, I suggest try larger or more complex datasets (such as ArXiv or Wiki) where deeper GNNs may be more beneficial.
4. Section 2.2 provides a clear explanation for straight-line graphs. However, graphs often contain more complex patterns. To strengthen your argument, I suggest to synthesize graphs with various patterns and run experiments on them. You may consider the methodologies of [1] and [2]. This would provide strong evidence that HGAT generalizes well across different graph topologies, node counts, sparsity levels and other conditions.
5. GCN, GraphSAGE, and GAT do not support edge weights or edge tensors. Whether your method can be applied to models that contains edge weights?

[1] Geisler, Simon, et al. "Transformers meet directed graphs." International conference on machine learning. PMLR, 2023.

[2] He, Yixuan, Gesine Reinert, and Mihai Cucuringu. "Digrac: Digraph clustering based on flow imbalance." Learning on Graphs Conference. PMLR, 2022.

---

> ### Author Response · Authors · 2025-08-31
>
> Thank you for your insightful comment. Below we provide our responses.
>
> **Response 1:** We acknowledge the importance of presenting numerical results for clarity and completeness. However, our primary goal with Figure 9 is to convey how model performance evolves with the number of layers, especially in the context of the K-skewed-traversal problem. We already mentioned this in the 2nd para. of Section 5.2 (page 11) of the manuscript.
>
> HGAT specifically addresses the K-skewed-traversal issue, which significantly affects performance in deeper GNN architectures. Consequently, the purpose of our experiments is to emphasize its resilience to performance degradation as the number of layers increases. Therefore, the plots in Figure 9 provide a more meaningful representation by illustrating how accuracy degrades with depth across different models. This better highlights HGAT's strength: a slower decline in performance, which cannot be effectively demonstrated with static, single-layer numerical snapshots. A static table of numerical results at a fixed number of layers would fail to capture this dynamic behavior, potentially obscuring the significance of HGAT's design in mitigating this problem.
>
>
>
> **Response 2:** We appreciate your careful examination of Figure 9. Our claim of HGAT’s superiority is not based on achieving the highest absolute accuracy at all depths. Rather, it is grounded in HGAT’s ability to sustain performance as the number of layers increases, which directly addresses the K-skewed-traversal problem. We already mentioned this in the 3rd para. of Section 5.2 (page 11) in the manuscript.
>
> Indeed, as you pointed out, baseline methods like GCN may slightly outperform HGAT in terms of absolute accuracy at shallow depths (e.g., 2–4 layers). However, these methods suffer sharp performance degradation beyond that point. In contrast, HGAT (both mean and sum variants) demonstrates a much more gradual decline in accuracy, maintaining higher relative performance at deeper layers.
>
> This trend is consistent across all three datasets (Cora, Citeseer, and Pubmed) and aligns with our core claim: HGAT mitigates the negative effects of stacking GNN layers — the K-skewed-traversal problem—resulting in more robust performance in deep GNN architectures.
>
>
>
> **Response 3:** Thank you for your insightful suggestion. As discussed in Section 5.2 of the manuscript, all baseline methods suffer from three core issues in deep architectures: over-smoothing, over-squashing, and the K-skewed-traversal problem. In contrast, HGAT is designed to address the K-skewed-traversal problem and therefore only suffers from the remaining two (over-smoothing and over-squashing).
>
> The intention of our current experiments is not to show that deep GNNs work well on these datasets in general, but to isolate and evaluate the impact of the K-skewed-traversal issue. Since this problem becomes more severe with increased depth, we analyze how performance degrades across layers for each method. The key observation is that HGAT degrades more gracefully, highlighting its ability to mitigate the K-skewed-traversal problem. (See Section 5.2 in the manuscript)
>
> That said, we agree that larger or more complex datasets (e.g., ArXiv, Wiki) may provide further insights into HGAT’s practical benefits in real-world deep GNN scenarios. Due to time constraints, we do not include it in this version. We consider that an important direction for future work and plan to extend our experiments accordingly.
>
>
>
> **Response 4:** We fully agree with your observation. The K-skewed-traversal problem is inherent to the aggregation mechanisms of traditional GNNs, and as demonstrated in our examples, it can arise even in simple topologies like straight-line graphs. Naturally, this issue persists—and often intensifies—in more complex graph structures due to their irregular neighborhood patterns. We have added this point in the revised manuscript (6th para. of Section 2.2.2, page 6). As per your suggestion, we include examples with more diversified and realistic graph structures in Appendix A (page 15-16) of revised manuscript to further illustrate the generality of the problem and strengthen our empirical justification.
>
>
> **Response 5:** Thank you for raising this important point. HGAT, in its current form, does not incorporate edge weights or edge tensors. This design choice was made to ensure a fair and consistent comparison with baseline methods such as GCN, GraphSAGE, and GAT, which also do not support edge weights in their standard implementations.
>
> We hope this clarifies the concerns raised, and we sincerely appreciate your time and effort in reviewing our work.

---

> > ### Comment · Reviewer_z97w · 2025-09-11
> >
> > Thank you for your response, which helped clarify my concerns.
> >
> > Regarding Weakness 1, I believe a detailed numerical table is necessary for a paper. You can create a table with certain key points. While graphs and paragraphs are useful for illustrating ideas, tables provide precise numerical results that are essential for clarity and reproducibility. In addition, such tables allow future researchers to directly compare their methods with yours, which strengthens the usability of your work.

---

> > > ### Author Response · Authors · 2025-09-14
> > >
> > > Thank you for your insightful comment. To improve clarity and reproducibility, we have added detailed numerical accuracy tables (Table 8, 9, 10) for all methods on the Cora, Citeseer, and Pubmed datasets in **Appendix B** of the revised manuscript. These tables complement the plots in Figures 9.
> > >
> > > We hope this addresses the concern raised and sincerely thank you for the time and effort you have dedicated to reviewing our work.

---

### Review · Reviewer_KXhS · 2025-06-16

**Summary Of Contributions:**

The authors analyse in GNNs how frequently the node representation of one node influences another during the GNN computation. Then they note this is skewed, in that nodes with a greater distances are integrated fewer than those closer by. To ensure this is equal, they design a new model that aggregates per-hop and then mixes between hops.

**Audience:**

Yes

**Claims And Evidence:**

No

**Requested Changes:**

**Critical:**

**Section 2**
- Why are specifically GCN, GraphSAGE and GAT chosen? It is not clear if GGNNF is broad enough to capture the wide range of GNN architectures studied in the literature.
- The definition of GGNNF is too imprecise. It is unclear what are the restrictions on the function 'AGGREGATE' and 'CONCAT'. It is not clear that CONCAT is not the actual concatenation function but something the different frameworks can instantiate (unless I'm missing some detail)
- The three examples do not directly instantiate AGGREGATE and CONCAT, making it unclear how the different architectures instantiate GGNNF.
- The fact that deterministic and randomised have two separate analyses suggests to me that the GGNNF framework is not precise enough to study the K-skewed traversal problem. For instance, what about using different sampling algorithms than GraphSAGE (Eg FastGCN or LADIES)? Or what about deterministic GNNs that somehow do not integrate certain nodes in their neighbourhood? Increasing this formalism would improve the paper and the analysis.
- Please define $H(k, v)$ more formally. It is unclear if this is the set of nodes where the shortest path to $v$ is length $k$, or the nodes that can be reached in $k$ hops. I assumed the former.
- (1) A high disparity in AIC can 'severely affect the GNNs ability to learn effective node representations'. This is a core claim, but not substantiated either theoretically or experimentally.
- It was not clear to me why the analysis of Randomized GGNNFs is included for the story of the paper.
- Page 6: Add a subsection header before 'Based on'
- The 3 propositions discussed at the end of Section 2 do not meet the bar of formal analysis. None have an actual proof attached, rather than just restating the proposition in a logical statement. Furthermore, it is unclear why involving modal logic using the $\square$ is helping the paper. It seems unnecessary here unless I'm missing something. Finally, it does not state any assumptions on the propositions, nor what the space of graphs is that's being considered. There certainly are graphs for which there are counterexamples to the propositions (eg proposition 1 when using a single-node graph).
- (2) Proposition 1: Please first formally define what 'prioritize' means. Just that the AIC values at different layers are different? I think prioritize is much too strong a word. Why could a GNN/GAT not learn to compensate for this discrepancy?
- Proposition 3: It is not immediately obvious from the examples why there is a monotonicity relation, so that needs a further formal proof.

**Section 3**
- This algorithm is a significant departure from the message passing framework and relies on being able to quickly compute k-hop distance for each pair of nodes. How is this implemented efficiently in practice? Can it be parallelised quickly? What is the computational complexity?
- I failed to understand figure 7. Shouldn't the labels be reversed?
- "HAG (...) eliminates the skew in hop contributions": Without weighting, you could say that each hop contributes equally. However, the individual nodes of further hops still contribute less than that of earlier hops, since usually earlier hops contain much fewer nodes than further nodes. Therefore nodes in further hops get summarised much more than in earlier hops. This aspect is not clearly discussed.
- "HGAT enhances the expressive power of GNNs, particularly in deep architectures": In what sense of the word 'expressive' is this true? This also requires a formal proof (and Theorem 1 is not proving this statement).

**Section 5**:
-  Table 6: Looking at the large standard deviations, assuming normality during reporting does not seem appropriate. Using quantiles is probably a better way to summarise the values.
- HGAT does not clearly improve performance-wise on the baselines, and the main result seems to be that the performance degrades less as the number of layers increase. This is not necessarily a compelling argument - Why would we want more layers if none of the methods do better then?
- It is not clear if the relative resilience to number of layers is because the method can learn to only put weight on the first few hops - effectively acting as having only a few layers. This aspect should be reported.
- There are no comparisons to methods handling oversmoothing and oversquashing. Therefore it is not clear whether the performance degradations of the baselines are because of K-Skewed Traversal or because of over smoothing/squashing. Combined with my points (1) and (2), I remain unconvinced that K-Skewed Traversal is an issue in practice.

**Improvements**
- Calling this 'attention' is diluting the word somewhat. Attention in Deep Learning architectures usually comes with key-query-value attention, and is input-dependent. This is just a weighted mixture of four embeddings.

**Strengths And Weaknesses:**

Strengths:
- Analysing how certain notes influence the final GNN decision is a relevant and important question for future understanding and algorithmic improvements of GNNs.
- The storyline of the paper is easy to follow.

Weaknesses:
- The experimental evaluation is very limited and does not convince me that 'k-skewed-travel' is actually a problem in practice.
- The formal analysis is lacking in precision.
- The paper is not well integrated with the existing literature, with little discussion of related (recent) work. However, I'm not an expert on the literature of GNN architecture design.
- The proposed solution is not clearly explained, and leaves me wondering about several important questions for deploying the architecture.

---

> ### Author Response · Authors · 2025-08-31
>
> Thank you for your insightful comment. Please find the specific responses to your comments.
>
> **Response to Summary Of Contributions:** The K-skewed-traversal manifests in three ways as specified in Proposition 1, Proposition 2, and Proposition 3 in Section 2. Proposition 1 says that depending on the subgraph structures, GNNs can assign different priorities to different nodes. For instance, a GNN might prioritize a node’s second-hop neighbors over its first-hop neighbors, while another node might experience the opposite priority. We already mentioned this in the manuscript (4th para. of Section 1, page 2).
>
> **Response to Weaknesses:** In the abstract (page 1), we specified that Graph neural networks (GNNs) often suffer performance degradation as their layer count grows, typically due to the well-known problems of over-smoothing and over-squashing. In this work, we identify an additional factor contributing to this degradation, which we term the K-skewed-traversal problem. The over-smoothing and over-squashing are well discussed in the literature, but, as we have newly identified the K-skewed-traversal problem, there is no related work discussing it.
>
> # Responses to Section 2 comments:
> **Response 1:** As mentioned in Section 1 (see para. 2 of Section 1, page 1) in the revised manuscript, GCN, GAT, and GraphSAGE are the pioneers of the message-passing graph neural network: a node integrates nearby nodes’ representations. Most of the other methods in the message-passing graph neural network are derived from these methods and use a message-passing framework. This is the reason that we experimented with these methods. Additionally, they are used as the backbone of some GNN methods [1].
>
> [1] You, Yuning et. al.“Graph Contrastive Learning with Augmentations.” NeurIPS, 2020.
>
> **Response 2 and 3:** GraphSAGE defines AGGREGATE using several choices—mean, LSTM, and pooling aggregators—and uses literal vector concatenation for CONCAT. In GGNNF, we abstract AGGREGATE and CONCAT to general placeholders, not tied to specific operations. The goal is to offer a unified structural abstraction over standard GNNs like GCN, GraphSAGE, and GAT, focusing on how node representations are recursively built over multiple hops.
>
> The examples in Sections 2.2.1 and 2.2.2 demonstrate which nodes are structurally consulted in forming a target node’s representation, regardless of how their features are combined. While the actual instantiations of AGGREGATE and CONCAT vary across models, our analysis is independent of them. What matters is the computation graph itself—i.e., which nodes appear in the aggregation paths and how frequently. This defines each node’s structural contribution, which we quantify using the Average Integration Count (AIC).
>
> Therefore, even without concretely specifying AGGREGATE and CONCAT, the imbalance in contribution across hops can be meaningfully analyzed, revealing the K-skewed-traversal problem. We have addressed this point in the revised manuscript (2nd para. of Section 2.1, page 2; 1st para. of Section 2.2.1, page 4).
>
> **Response 4:** The K-skewed-traversal problem occurs where recursive neighborhood expansion across layers is used: each node’s representation is generated by integrating information from its neighbors’ previous representations and concatenating this with its own previous representation. (We have already stated this in the 2nd para. of Section 1, page 1 of the manuscript). This can be done deterministically or randomized. FastGCN and LADIES are out of the scope of our proposal because recursive neighborhood expansion across layers is not used. Additionally, the recursive neighborhood integration can occur in one of two ways: deterministic and randomized, causing us to analyze them separately. Importantly, despite differences in neighborhood integration, both ultimately lead to the same consequence: the K-skewed-traversal problem (stated in 1st para. of Section 2.2, page 3 of the manuscript; clarified further in the revised version in 1st para. of Section 2.2, page 4).
>
> **Response 5:** $H(k, v)$ means the set of nodes where the shortest path to $v$ is length $k$, equivalently the set of nodes that can be reached in hops $k$ from node $v$.
>
> Due to character count limitations, we provide the remaining responses in the following continuation.

---

> > ### Author Response · Authors · 2025-08-31
> >
> > (Continuation from previous comment)
> >
> > **Response 6:** The K-skewed-traversal problem results in a high disparity in Average Integration Count (AIC). As shown in Sections 2.2.1 and 2.2.2, along with Proposition 3 and Table 7, AIC disparity increases with depth, leading to irregular and uneven contributions from different hops during message aggregation.
> >
> > Empirically, Section 5.2 and Figure 9 demonstrate that as the number of layers increases, baseline models—GCN, GraphSAGE, and GAT—exhibit significant performance degradation on benchmark datasets, while HGAT remains comparatively stable. Table 6, 7 shows the imbalances of AIC values on the benchmark datasets–Cora, Citeseer, Pubmed–in accordance with Proposition 1, 2, 3. Since HGAT is specifically designed to eliminate AIC disparity (as established in Theorem 1), this performance contrast isolates and supports the impact of the K-skewed-traversal problem.
> >
> > In other words, the sharp performance drop in baseline models—corresponding to increasing AIC disparity—substantiates our claim that high disparity in AIC can severely impair a GNN’s ability to learn effective node representations.
> >
> > **Response 7:** The K-skewed-traversal problem arises in any framework using recursive neighborhood expansion, whether the aggregation is deterministic or randomized. To provide a complete understanding of how this problem manifests across GNN variants, we analyze both deterministic and randomized GGNNFs. This ensures our findings generalize across different model designs, such as GraphSAGE (randomized) and GCN (deterministic). or randomized. (stated in 1st para. of Section 2.2, page 3 of the manuscript; clarified further in the revised version in 1st para. of Section 2.2, page 4).
> >
> >
> > **Response 8:** Thank you for the suggestion. Since the propositions are part of the explanation of the K-skewed-traversal problem, and adding more subsection headers may fragment the narrative, we chose to keep them under the same section.
> >
> > **Response 9:** The use of modal logic is intended to express the generality of the observed behaviors across nodes, helping to formalize the K-skewed-traversal problem. We found it a compact and expressive tool.
> >
> > The propositions are empirical observations supported by results in Section 5 (e.g., Tables 6, 7, and Figure 9).
> >
> > Regarding Proposition 1, the reviewer’s single-node graph example is not a valid counterexample, as the proposition needs two distinct hop levels $k_1 \ne k_2$, which a single-node graph does not satisfy.
> >
> > **Response 10:** Prioritizing means a node is being more integrated in the target node’s embedding than another node, this can be quantified by AIC (the definition of AIC is provided in Section 2.2.1.)
> >
> > As we stated in Section 2, the K-skewed-traversal problem is not a deliberate design choice, but rather an unintended byproduct of the standard aggregation process in models like GCN, GraphSAGE, and GAT. For example, in GCN and GraphSAGE, this problem arises from unintentional biases in their aggregation processes. In GAT, which introduces attention, the issue manifests even before the attention parameters have been learned, since the initial node representations are already skewed when the attention weights are applied. These biases are not explicitly encoded rules or optimizations; rather, they intrinsically arise from the normal computation in traditional GNNs. This is already stated in 5th para. of Section 1, page 2 of the manuscript.
> >
> > **Response 11:** Thank you for pointing this out. Proposition 3 states that in GGNNF, adding more layers does not decrease the Average Integration Count (AIC). While the examples in Section 2.2 illustrate this behavior, and the empirical results in Section 5 (Tables 6, 7, and Figure 9) support it.
> >
> > # Responses to Section 3 comments:
> > **Response 12:** Thank you for this important question. In our implementation, $k$-hop neighborhoods are computed using breadth-first search (BFS) (or depth-limited DFS) up to a depth of $k$. The time complexity for computing the $k$-hop neighborhood of a node is $\frac{D^{k+1}-1}{D-1}$ where $D$ is the average node degree.
> >
> > This process is easily parallelizable, since the $k$-hop neighborhood of each node can be computed independently of others. In practice, we precompute these neighborhoods before model training, ensuring scalability for moderately sized graphs. We have added this point in the revised manuscript (7th para. of Section 3.1, page 9).
> >
> > **Response 13:** Thank you for pointing this out. You are correct — the labels in Figure 7 were inadvertently reversed. We have corrected it in the revised version (see figure 7, page 8).
> >
> > Due to character count limitations, we provide the remaining responses in the following continuation.

---

> > > ### Author Response · Authors · 2025-08-31
> > >
> > > (Continuation from previous comment)
> > >
> > > **Response 14:** You raise an important point. The phenomenon you describe — where many distant nodes are aggregated into a compact summary, resulting in loss of information — is known as over-squashing. As already discussed in 3rd para. of Section 1 of the manuscript, this occurs because the number of contributing nodes grows exponentially with depth, but their information is compressed into a fixed-size representation.
> > >
> > > However, our contribution, the K-skewed-traversal problem, identifies an additional and orthogonal factor: even at the same hop level, different nodes may receive disproportionate contributions from that hop depending on sub-graph structure. This leads to random hop-level prioritization across nodes, which is not captured by over-smoothing or over-squashing alone.
> > >
> > > HGAT specifically targets and resolves the K-skewed-traversal problem by enforcing uniform hop-level aggregation before applying attention. It does not attempt to resolve over-squashing, which remains a separate challenge. This is central to our contribution. These are central to our contribution, discussed and established throughout the paper.
> > >
> > > **Response 15:** Thank you for pointing this out. By “expressive,” we refer to HGAT’s improved ability to preserve and utilize multi-hop information in deep architectures, as shown by its resistance to performance degradation compared to baselines (Figure 9, page 12).
> > >
> > > # Responses to Section 5 comments:
> > > **Response 16:** As stated in Section 5.1, we report the mean and standard deviation of AIC to summarize the values across nodes—not under the assumption of normality, but due to the impracticality of inspecting node-level AICs individually. These summaries are essential for validating our three propositions as done in the last two paragraphs of section 5.1 (page 11), which analyze how AIC behaves across hops and layers.
> > >
> > > While we acknowledge that large standard deviations suggest non-normality, quartile statistics alone cannot capture the behavior of AIC—which is central to our findings in Propositions 1, 2 and 3 (see section 5.1). These propositions rely on expected integration growth across hops and layers, which is best reflected through mean-based analysis. Therefore, we retain mean and standard deviation as the most appropriate and informative metrics for our analysis.
> > >
> > >
> > > **Response 17:** Performance degradation with increased depth is a well-known issue in GNNs, typically attributed to over-smoothing and over-squashing. Our contribution identifies the K-skewed-traversal problem as an additional cause, offering a new perspective on why deeper GNNs often fail. While deeper layers may not improve performance on current benchmarks, understanding and isolating this problem enhances our broader understanding of GNN behavior and opens the door to future architectures that can scale more effectively with depth.
> > >
> > > **Response 18:** The hop-wise attention weights remain distributed across multiple hops even in deeper networks.
> > >
> > > **Response 19:** As discussed in Section 5.2, all baseline methods suffer from three core issues in deep architectures: over-smoothing, over-squashing, and the K-skewed-traversal problem. In contrast, HGAT is designed to address the K-skewed-traversal problem, and therefore only suffers from the remaining two (over-smoothing and over-squashing). That way we isolate and evaluate the impact of the K-skewed-traversal issue as shown in section 5.2. Since this problem becomes more severe with increased depth, we analyze how performance degrades across layers for each method. The key observation is that HGAT degrades more gracefully, highlighting its ability to mitigate the K-skewed-traversal problem. This is already discussed in Section 5.2.
> > >
> > > # Response to improvements comments:
> > > **Response 20:** Our use of the term “attention” follows the precedent set by Graph Attention Networks (GAT), where attention is implemented as a weighted sum without explicit key-query-value mechanisms. Similarly, our method computes learned hop-wise weights based on input features, aligning with this established usage.
> > >
> > > We hope this clarifies the concerns raised, and we sincerely appreciate your time and effort in reviewing our work.

---

> > > > ### Comment · Reviewer_KXhS · 2025-09-01
> > > >
> > > > I thank the authors for their extensive answer, which clarified several of my concerns and a few misconceptions.
> > > >
> > > > I still have a few open /followup questions
> > > > - Do the authors believe that increasing layer counts is still an issue for methods that resolve oversquashing/smoothing? That is, is k-skewness entirely orthogonal to these two dimensions?
> > > >     - I believe this point would be a lot clearer if the authors also included a baseline that directly addressed oversquashing/smoothing to better understand how to disentangle these factors.
> > > >     - Also RE response 19. I find it somewhat hard to understand from this evidence whether k-skewness has been entirely isolated, as the authors claim in this response. While the method addresses it, it is also just quite a different model.
> > > > - Response 4: "FastGCN and LADIES are out of the scope of our proposal because recursive neighborhood expansion across layers is not used." As far as I understand, these two methods do use recursive neighbourhood expansion. Maybe I'm missing the point.
> > > > - Response 5: This is not updated in the manuscript.
> > > > - Response 6: So do I understand correctly that the backing of this claim is primarily figure 9?
> > > > - Response 9: A proposition in an ML paper refers to something for which one has a formal proof. The authors however here argue it's just an empirical observation. Therefore, these should not be propositions. (Furthermore, the lack of assumptions is still not handled in the revised version).
> > > > - Response 11: This is again insufficient for a formal claim, as there could be counterexamples that are missed.
> > > > - Response 16: This did not really resolve my question. It is not clear to me why quartiles (or any related such method) are not appropriate here.
> > > > - Response 18: This should be part of the paper, IMO. As it would be an easy way out for the model to only assign to early layers.

---

> > > > > ### Author Response · Authors · 2025-09-14
> > > > >
> > > > > Thank you for your insightful comment. Please find the specific responses to your comments.
> > > > >
> > > > > **Regarding first follow-up questions**: As stated in Response 19 and discussed in Section 5.2, all baseline methods (GCN, GraphSAGE, GAT) suffer from three major issues in deep architectures: over-smoothing, over-squashing, and the K-skewed-traversal problem. In contrast, HGAT is specifically designed to eliminate the K-skewed-traversal problem, leaving only the first two issues (over-smoothing and over-squashing). This distinction is summarized in the table below and allows us to isolate and evaluate the specific impact of K-skewness. Since skewness becomes more severe with depth, we analyze performance degradation across layers and find that HGAT degrades more gracefully, highlighting its effectiveness in mitigating this issue (Section 5.2).
> > > > >
> > > > > | Methods| Issues present|
> > > > > |--------|-----|
> > > > > | GCN, GraphSAGE, GAT| Over-smoothing, Over-squashing, **the K-skewed-traversal problem**|
> > > > > | HGAT-mean, HGAT-sum| Over-smoothing, Over-squashing|
> > > > >
> > > > > We emphasize that these issues are **orthogonal by definition**: their mechanisms differ, even though they all intensify with increasing depth. Any method based on recursive neighborhood expansion—where each node’s representation is built from neighbors’ previous representations combined with its own—will inherently face the K-skewed-traversal problem, even if it also addresses over-smoothing or over-squashing (see para. 4-5 of Section 1, page 2).
> > > > >
> > > > > We agree that including baselines designed to directly mitigate over-smoothing or over-squashing would help disentangle these factors more clearly. We did not include them in this version. We regard it as an important direction for future work, though it is apparent that the main message of our contribution would not be affected by this.
> > > > >
> > > > >
> > > > >
> > > > > **Regarding Response 4**:
> > > > > After close inspection of FastGCN and LADIES, we observe:
> > > > > **FastGCN** samples nodes from a global pool, biased by degree, rather than strictly from the local neighborhood. If sampled nodes are not actual neighbors of the target node, their contributions vanish.
> > > > > **LADIES** performs layer-wise sampling: nodes are sampled only from the neighbors of already-selected nodes at the next layer.
> > > > >
> > > > >
> > > > > Thus, while FastGCN’s sampling differs from GraphSAGE, LADIES is more closely related. Both methods are susceptible to the K-skewed-traversal problem. We agree that adding these baselines enhances our contribution. We did not include them in this version. We consider this an important direction for future work, though it is apparent that the main message of our contribution would not be affected by this.
> > > > >
> > > > > **Regarding Response 5**:
> > > > > We have added the definition of $H(k, v)$ in the revised manuscript. (2nd para. of Section 2.2.1, page 4).
> > > > >
> > > > >
> > > > > **Regarding Response 6**:
> > > > > We first introduce the K-skewed-traversal problem. We then introduce AIC to provide a quantitative measure of the K-skewed-traversal problem. Section 2 and 5.1 analyze how skewness manifests through AIC, and HGAT is then proposed to directly resolve this imbalance. Empirical validation (Figure 9) demonstrates that benchmarks confirm the link between high AIC disparity and performance degradation. Thus, the claim is not supported by Figure 9 alone—the entire paper is connected: AIC quantizes the K-skewed-traversal problem, HGAT addresses it, and empirical evaluation substantiates it.
> > > > >
> > > > >
> > > > > **Regarding Response 9, 11**:
> > > > > Defining the K-skewed-traversal problem using plain language alone can be imprecise. To reduce ambiguity, we supplement the explanation with formal logical expressions, providing precise formulations of the observed behaviors. These formulations are supported by examples and experiments (e.g., Appendix A, Tables 6–7, Figure 9). Our goal is clarity, and the propositions are designed to fully capture the K-skewed-traversal problem.
> > > > >
> > > > > The propositions are expressed to comprehensively characterize the K-skewed-traversal problem. If any assumptions are claimed to be missing, they should be explicitly stated. Similarly, any counterexamples to Proposition 3 should include at least one concrete instance.
> > > > >
> > > > >
> > > > > Due to character count limitations, we provide the remaining responses in the following continuation.

---

> > > > > > ### Author Response · Authors · 2025-09-14
> > > > > >
> > > > > > (Continuation from previous comment)
> > > > > >
> > > > > > **Regarding Response 16**:
> > > > > >
> > > > > > As discussed in Section 5.1, we report the mean and standard deviation of AIC across nodes. This choice is not based on an assumption of normality but on the practical need for aggregate summaries, since inspecting node-level AICs individually is impractical. These statistics are essential for validating Propositions 1–3 (Section 5.1, last two paragraphs, page 11).
> > > > > >
> > > > > > While quartile-based statistics (median, Q1, Q3, IQR) capture rank-based spread, they do not preserve the additive structure of AIC, which is central to our analysis:
> > > > > >
> > > > > > - **Mean:** reflects cumulative contributions (sum divided by count).
> > > > > > - **Variance/standard deviation:** measures squared deviations from the mean and is also additive.
> > > > > >
> > > > > >
> > > > > > Because our analysis (last two para. of Section 5.1) relies on additive expectations of integration counts, quartiles alone cannot capture the growth dynamics of AIC with increasing K. In short, mean and standard deviation allow us to analyze the imbalance in cumulative contributions across hops, whereas quartiles only describe distribution spread. Therefore, mean and standard deviation remain the most appropriate metrics for our analysis.
> > > > > >
> > > > > > Specifically, Propositions 1–3 are supported using mean-based statistics in the last two paragraphs of Section 5.1. In Table 6, the varying mean of AIC illustrates that AICs across different hops are not necessarily equal, as stated in Proposition 1. In Table 7, the increasing mean of AIC demonstrates that adding layers to the GGNNF increases the AIC of a node’s specific hop, as stated in Proposition 3. Such effects cannot be inferred from the median or other quartile statistics.
> > > > > >
> > > > > > **Regarding Response 18**:
> > > > > > We already stated in Section 3.1 (para. name: Hop-wise Attention Mechanism (Phase 2), page 8) that HGAT combines hop-wise summaries using attention weights learned during training. If the model were to assign weight only to early hops—effectively reducing itself to shallow layers—its performance should plateau after a few layers. However, Section 5.2 and Figure 9 (pages 11-12) show that as the number of layers increases, all models (GCN, GraphSAGE, GAT, HGAT) degrade. Yet, while baselines suffer sharp declines, HGAT (especially the mean variant) degrades more gradually, maintaining stronger performance at greater depths. This demonstrates that HGAT does not collapse into **shallow behavior** and instead achieves depth robustness.
> > > > > >
> > > > > > We hope this addresses the concerns raised and sincerely thank you for the time and effort you have dedicated to reviewing our work.

---

### Review · Reviewer_cneE · 2025-08-17

**Summary Of Contributions:**

The paper introduces the Generalized GNN Framework that unifies deterministic and randomized message-passing models for multi-hop analysis. It formalizes the K-skewed-traversal problem via the Average Integration Count, showing cross-hop imbalance that depends on local structure and increases with depth. It proposes the Hop-wise Graph Attention Network (HGAT) with uniform per-hop aggregation followed by hop-wise attention, proves Phase 1 yields AIC = 1, and reports improved depth robustness on citation benchmarks against GCN, GraphSAGE, and GAT.

**Audience:**

Yes

**Claims And Evidence:**

No

**Requested Changes:**

Please refer to the weakness part.

**Strengths And Weaknesses:**

Strengths:

1. GGNNF unifies GCN, GraphSAGE, and GAT in a single message passing template and enables consistent multi hop analysis for deterministic and randomized models.
2. AIC formalizes the K-skewed-traversal problem by quantifying cross hop integration imbalance and shows it depends on local structure and increases with depth.
3.HGAT applies uniform per hop aggregation followed by hop wise attention, Phase 1 guarantees AIC=1 and experiments show improved depth robustness over GCN, GraphSAGE, and GAT.

Weaknesses:

1. The method reduces to a learnable weighting over {A^k X}, closely related to multi-order/multi-hop mixing (e.g., GPR-GNN, MixHop). Without head-to-head comparisons or a formal theoretical distinction from these baselines, the contribution appears incremental.

2. The problem formulation lacks a precise objective and success criterion. It is unclear whether the goal is to enforce AIC=1 or merely reduce imbalance. There is no explicit loss or constraint linking AIC to training. The goal–method mapping remains implicit, and key assumptions and scalability targets are unstated.

3. The evaluation is limited in scope. Benchmarks are small citation graphs and the task is node classification only. Strong related baselines are missing, notably GPR-GNN and MixHop. Core ablations are absent, including Phase-1 only, Phase-2 only, K sensitivity, aggregator choice, and shared versus node-dependent attention. Statistical reporting is weak, with no multi-seed means or confidence intervals and unclear hyperparameter fairness or compute parity. Complexity is not analyzed, with no runtime or memory profiling versus K or graph size, and no large-scale scalability study.

---

> ### Author Response · Authors · 2025-08-31
>
> Thank you for your insightful comment. Please find the specific responses to your comments.
>
> **Response 1:** The K-skewed-traversal problem arises in models such as GCN, GraphSAGE, and GAT, where each layer builds node representations by repeatedly integrating neighbors’ embeddings from the previous layer, leading to skewed hop contributions. GPR-GNN and MixHop also incorporate multi-hop signals, but they do so by propagating previous-layer embeddings from all hops up to K simultaneously. This design actually intensifies the K-skewed-traversal problem, as multiple-hop neighbors are recursively integrated at once. HGAT differs fundamentally: it aggregates raw features from nodes up to K hops (not recursively propagated embeddings), ensures uniform per-hop integration ($AIC = 1$), and then applies hop-wise attention.
> Due to time constraints, we do not include GPR-GNN and MixHop in this version. However, we acknowledge their relevance and view head-to-head comparisons and deeper theoretical distinctions as an important direction for future work.
>
> **Response 2:** The objective of this paper is to directly address the K-skewed-traversal problem in GGNNF by enforcing uniform contributions from all hop distances when constructing node representations, ensuring an Average Integration Count (AIC) of 1 across layers. Here, $AIC = 1$ indicates balanced hop contributions, whereas skewed AIC deviations correspond to skewed contributions, i.e., the K-skewed-traversal problem. Theorem 1 (Section 4, page 9-10) formally proves that HGAT achieves $AIC = 1$ in Phase 1, while Section 2 illustrates how other methods yield skewed AIC values.
> As discussed in Section 5.2 of the manuscript, baseline models (GCN, GraphSAGE, GAT) degrade with depth due to over-smoothing, over-squashing, and K-skewed-traversal (imbalanced AIC). HGAT eliminates the third issue by construction, isolating its effect. Empirically, since the problem intensifies with depth, we compare performance degradation across layers. HGAT degrades more gracefully, confirming its effectiveness in mitigating the K-skewed-traversal problem.
>
> **Response 3:**
>
> *Regarding usage of citation network on node classification task:* We followed prior state-of-the-art work (GCN, GAT) in evaluating citation benchmarks for node classification, which are standard in the literature. While we agree that larger-scale or more diverse tasks (e.g., link prediction, heterophily benchmarks) would provide additional insights, these are left as future extensions due to time limitation.
>
> *Regarding missing baselines:* As noted in Response 1, MixHop and GPR-GNN exacerbate K-skewed-traversal even more than the models we tested. Including them requires theoretical extension and extensive experiments, which we could not accommodate here due to time limitation. We plan to extend HGAT evaluation to these models in future work.
>
> *Regarding ablation study:* Since Phase 2 operates on top of the uniformly aggregated outputs of Phase 1, it cannot function independently. Therefore, we can argue that HGAT is atomic on the two phases: Phase 1 and Phase 2.
>
> *Regarding K sensitivity:* The entire paper analyzes how performance changes with increasing K, which is central to our contributions.
>
> *Regarding aggregator choice:* As we have stated in the first paragraph of Section 3.1 page 9 of the manuscript, We used mean/sum for HGAT and GraphSAGE as they are differentiable and standard. GraphSAGE itself defines AGGREGATE as mean, pooling, or LSTM. In GGNNF, AGGREGATE and CONCAT are abstract placeholders, since our analysis depends only on the structural computation graph, not the exact combination rule. The examples in Sections 2.2.1 and 2.2.2 demonstrate which nodes are structurally consulted in forming a target node’s representation, regardless of how their features are combined. While the actual instantiations of AGGREGATE vary across models, our analysis is independent of them. What matters is the computation graph itself—i.e., which nodes appear in the aggregation paths and how frequently. We have addressed this point in the revised manuscript (2nd para. of Section 2.1, page 2-3; 1st para. of  Section 2.2.1, page 4).
>
> *Regarding attention type:* The title of our paper is “Addressing Node Integration Skewness in Graph Neural Networks Using Hop-Wise Attention”, which clarifies that the attention is hop dependent, not node.
>
> *Regarding hyperparameter fairness:* We optimized hyperparameters separately for each method, trained for 600 epochs at every K because training accuracy plateaus beyond that, and reported the best test accuracy observed across all epochs. We have addressed this point in the revised manuscript (1st para. of Section 5.2, page 11).
>
> Due to character count limitations, we provide the remaining responses in the following continuation.

---

> > ### Author Response · Authors · 2025-08-31
> >
> > **(Responses continued from previous comment)**
> >
> > *Regarding complexity:* In our implementation, $k$-hop neighborhoods are computed using breadth-first search (BFS) (or depth-limited DFS) up to a depth of $k$. The time complexity for computing the $k$-hop neighborhood of a node is $\frac{D^{k+1}-1}{D-1}$ where $D$ is the average node degree.
> >
> > This process is easily parallelizable, since the k-hop neighborhood of each node can be computed independently of others. In practice, we precompute these neighborhoods before model training, ensuring scalability for moderately sized graphs. We have added this point in the revised manuscript (7th para. of Section 3.1, page 9)
> >
> > *Regarding no statistical reporting with multi-seed means or confidence intervals, no runtime or memory profiling versus K or graph size, and no large-scale scalability study:* We agree that these will enhance our contribution. Due to time limitations, we did not do it. Consequently, in this paper we try to establish our idea as a contribution. We consider that an important direction for future work.
> >
> > We hope this clarifies the concerns raised, and we sincerely appreciate your time and effort in reviewing our work.

---

### Comment · Reviewer_z97w · 2025-06-05

**To keep safe, this comment is only visible to the author, TMLR Editors-in-Chief, and TMLR Paper 4763 Action Editors.**'

I notice that you upload the entire project folder, which helps demonstrate that your project is well-prepared. However, the folder also includes some temporary files containing personal information. This leads to an unintended information leak. Please carefully review the uploaded files and remove any unnecessary files as soon as possible.

---

> ### Author Response · Authors · 2025-08-30
>
> Thank you for your careful observation. We have reviewed the uploaded files and removed all unintended temporary and configuration files that may contain personal information.

---

### Decision · Action_Editor_dCxa · 2025-09-25

**Recommendation:** Reject

**Audience:**

Yes

**Audience Explanation:**

This is could attract some individuals in graph machine learning community.

**Claims And Evidence:**

No

**Claims Explanation:**

The paper introduces Average Integration Count to characterize cross-hop imbalance (K-skewed traversal) and proposes Hop-wise Graph Attention Network model. The idea is interesting and some results are promising. However, reviewers find the theory underdeveloped and the novelty incremental without formal separation from prior methods. Experiments lack key ablations, baseline methods, scalability/runtime analyses, and evaluations on larger datasets. Overall, reviewers agree that the current manuscript is not suitable for TMLR.